



# How extreme apparitions of the volcanic and anthropogenic south east Asian aerosol plume trigger and sustain: El Niño and Indian Ocean Dipole events; and drought in south eastern Australia. First attribution and mechanism using Global Volcanism Program, Last Millennium Ensemble, MERRA-2 reanalysis and NASA satellite data.

**Keith Alan Potts[1]**

[1]Kyna Keju Pty Ltd, Adelaide, 5041, Australia

*Correspondence to*: Keith A. Potts (Keith.Potts@bigpond.com)

**Funding:** This research project was funded personally by the author and his wife Julie.

**Summary**

El Niño events, the greatest inter-annual variation in the global climate, are and always have been caused by volcanic 15 aerosols in south east Asia. Recently the volcanic aerosols have been augmented by anthropogenic aerosols especially from September to November which has intensified ENSO events. The same aerosol plume also creates drought in south eastern Australia and Indian Ocean Dipole events simultaneously. Volcanic, modelling, reanalysis and measured data all show the same results.





**Abstract.**

Volcanic aerosols over south east Asia (SEAsia), and only over SEAsia, have always been the trigger and sustaining cause of: El Niño/Southern Oscillation (ENSO) events which are the dominant mode of variability in the global climate responsible for Australian, Indian and Indonesian droughts, American floods and increased global temperatures; and Indian

Ocean Dipole (IOD) events. In recent decades this natural plume has been augmented by an anthropogenic plume which has intensified ENSO events especially from September to November. Understanding the mechanism which enables aerosols over SEAsia, and only over SEAsia, to create ENSO events is crucial to understanding the global climate. I show that the SEAsian aerosol plume causes ENSO events by: reflecting/absorbing solar radiation which warms the upper troposphere; and reducing surface radiation which cools the surface under the plume. This inversion reduces convection in SEAsia

thereby suppressing the Walker Circulation and the Trade Winds which causes the SST to rise in the central Pacific Ocean and creates convection there. This further weakens/reverses the Walker Circulation driving the climate into an ENSO state which is maintained until the SEAsian aerosols dissipate and the climate system relaxes into a non-ENSO state. Data from the Global Volcanism Program (151 years), the Last Millennium Ensemble (1,156 years), MERRA-2 (41 years) and NASA MODIS on Terra (21 years) demonstrates this connection with the Nino 3.4 and 1+2 SST, the Southern Oscillation Index,

and two events commonly associated with ENSO: drought in south eastern Australia; and the IOD.





# 1    Introduction

South East Asia (SEAsia) is a unique region with several attributes which enable SEAsia, and only SEAsia, to create and sustain El Niño-Southern Oscillation Index (SOI) (ENSO) events and their associated effects such as drought in South Eastern Australia (SEAus) and Indian Ocean Dipole (IOD) events. SEAsia is a region which covers about 3% of the global surface and: hosted 26% of the eruptions in the Global Volcanism Program (GVP) database from 1800 to 2020 (Appendix A); drives the "non-ENSO" Walker Circulation; is the largest area of tropical convection in the world; drives the northern and southern regional Hadley Circulation; covers the eastern part of the Indian Ocean used to calculate the IOD; and in recent decades has hosted one of the eight extreme, anthropogenic, continental scale, aerosol plumes from September to November (SON) (Appendix B).

I demonstrate that volcanoes in SEAsia have always created and sustained ENSO events in three stages by showing:

**First** that volcanic tephra from the SEAsian eruptions has a statistically significant connection to the three ENSO indices, Nino 3.4 and 1+2 Sea Surface Temperature (SST), the SOI, and other ENSO related events using GVP data;

**Second** that aerosols over SEAsia are intimately connected to the three ENSO indices, using the Last Millennium Ensemble (LME) (Otto-Bliesner et al., 2016), MERRA-2 (Gelaro et al., 2017) and the NASA Terra/MODIS (Kaufman et al., 2000) and NCEP (Kalnay et al., 1996) (together TN) data; and

**Third** that the connection must flow from the aerosols to ENSO.

ENSO events are created in the following sequential stages:

1.    The South East Asian aerosol Plume (SEAP) is established;

2.    The SEAP aerosols absorb (and reflect) solar radiation which: (a) heats the atmosphere; and (b) reduces solar radiation at the surface under the plume which cools the surface; this

3.    Creates a temperature inversion compared to times without a plume which reduces convection;

4.    Reduced convection over SEAsia causes the Trade Winds (TW) blowing from east to west over the Pacific Ocean as part of the Walker Circulation to reduce in intensity as there is a reduced exit into the convection and the Walker/Hadley Circulation;

5.    Reduced TW speed over the Pacific Ocean causes the sea surface temperature to rise; which

6.    Creates convection in this region and the Walker circulation further relaxes or even reverses;

7.    The SOI is forced into a negative phase by these changes in convection;

8.    The western Pacific warm pool then migrates east as the wind stress on the ocean has reduced;

9.    The ENSO event continues until the aerosol plume over SEAsia dissipates which is typically when the SEAsian monsoon starts and washes the aerosols out of the atmosphere; and

10.    With the revived TW speed over the Pacific Ocean the warm pool then migrates west again and the ENSO event ends.

This paper follows this sequence to establish the connection between each step and the SEAP.





## 1.1 Areas used

The areas used in this analysis are shown in Fig. 1 and are:

1. The area covered by the SEAP (SEAP Area) 10º S-10º N and 90º E-160º E;

2. The Central SEAP (CSEAP) Area 5º S-5º N and 100º E-120º E;

3. The IOD Areas

4. SEAus pressure 130º to 145º E 25º to 40º S and Melbourne;

5. SEAus rainfall 137º to 160º E 30º to 40º S and Melbourne;

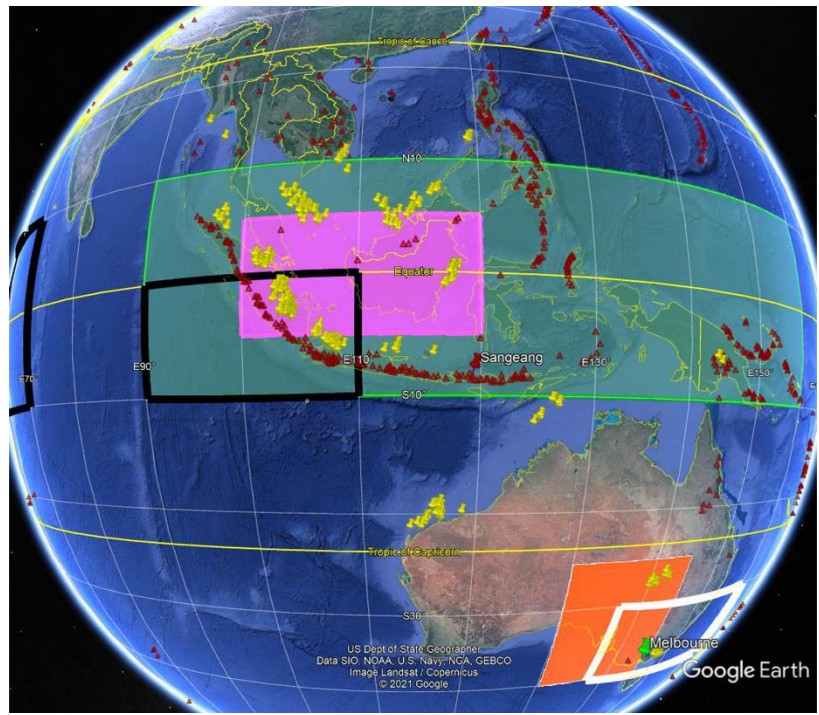

Figure 1: The SEAP Area, green; CSEAP Area, pink; IOD Areas, black outline; SEAus pressure, orange; SEAus rainfall, white outline; Melbourne, green pin; volcano locations from the GVP in red; and gas flares (NOAA and the Global Gas Flaring Reduction Partnership (GGFRP), yellow. Map data: Google Earth, US Dept of State Geographer, SIO, NOAA, U.S. Navy NGA, GEBCO; image Landsat/Copernicus. © Google Maps 2021.

## 1.2 ENSO

ENSO is the dominant interannual climate signal originating in the tropical Pacific and is driven by interactions between the atmosphere and ocean (McPhaden et al., 2006). It is defined in the Intergovernmental Panel on Climate Change (IPCC) Assessment Report Four (AR4) Glossary (Solomon et al., 2007) as a coupled atmosphere-ocean phenomenon with a two to





seven year time scale with significant effects on the climate in the Pacific region and across the world. ENSO events commence when the TWs blowing from east to west across the Pacific Ocean weaken (Enfield, 1989), McPhaden et al. (1998), (Brown and Fedorov, 2010), (Cai et al., 2015) and (Wang and An, 2002).

It is therefore clear that the TWs and therefore the Walker Circulation are intimately connected with ENSO events and determining the cause of the weakening of the TWs and Walker Circulation may well reveal the cause of ENSO events.

It is known that aerosol plumes can alter the major atmospheric circulation systems (Solomon et al., 2007), (Remer et al., 2009) and the SEAP is uniquely positioned to influence the Walker Circulation as it, and only it, exists in the region of non-ENSO Walker Circulation convection.

### 1.2.1 ENSO Theory and Models

Since (Bjerknes, 1969) first described the link between the atmospheric and oceanic components of ENSO events several theories have been developed to explain the triggering, development and decay of ENSO events. Bjerknes (1969) suggested that the ocean triggered ENSO events with the warmer SST creating convection in the central Pacific Ocean and slowing the Walker Circulation which created higher pressure in Djakarta/Singapore. However, this is at odds with the literature cited above which clearly states that relaxation of the Trade Winds triggers ENSO events. Kleeman and Moore (1997) suggest that the predictability of ENSO events is limited by the stochastic forcing of atmospheric transients and this is in accord with this paper which shows the atmospheric transients are created by the volcanic SEAP which is, of course, inherently unpredictable. Zebiak and Cane (1987) proposed a simple coupled atmosphere/ocean ENSO model which demonstrated oscillatory behaviour with a period of 3 to 4 years and stated that this oscillation is an internal characteristic of the coupled system as, apart from year one there is no external forcing. This may imply that the recurrence of ENSO in the LME data at a similar period across eight significantly different forcing scenarios is also an internal characteristic of the model. One inherent characteristic of ENSO events is their synchronisation to the annual cycle and Stein et al. (2014) suggest this is due to annual modulation of the coupled system. This paper demonstrates that the annual modulation is driven by the south east Asian monsoon which clears aerosols from the SEAP Area allowing convection and the TW's to be re-established and the oceanic elements of ENSO to relax back to their non-ENSO state over the following months. Dijkstra (2006) found that the internal variability and feedbacks within the coupled atmosphere/ocean system create ENSO events, whilst acknowledging that the mechanism which creates ENSO irregularity has not yet been identified which is what this paper does.

Wang (2018) reviewed the theories used to model ENSO and categorised them as: (1) a stable mode interacting with high frequency forcing; and (2) a self-sustaining oscillation which was split into four types: delayed; recharge-discharge; western Pacific; and advective-reflective. The paper concludes that there is no agreement yet on which category is correct.

This paper: supports (Kleeman and Moore, 1997) and (Dijkstra, 2006); demonstrates that the high frequency forcing is the volcanic tephra over SEAsia; and therefore shows that the category (2) oscillation theories are incorrect.





### 1.2.2    ENSO and volcanic eruptions

The anthropogenic SEAP has only existed in its current form since about 1980 (Fig. 2 which shows a 500% increase from 1980 to 2000 and Appendix B) and as ENSO events have most likely occurred for millions of years (NOAA at https://www.pmel.noaa.gov/elnino/faq) the driver of historic ENSO events must be the natural SEAP created by volcanic eruptions (Appendix A) and, according to the Volcano Global Risk Identification and Analysis Project at https://www2.bgs.ac.uk/vogripa/index.cfm, SEAP Area volcanoes have been erupting for the same period e.g. Toba's

earliest eruption is dated at 1.2 Ma.

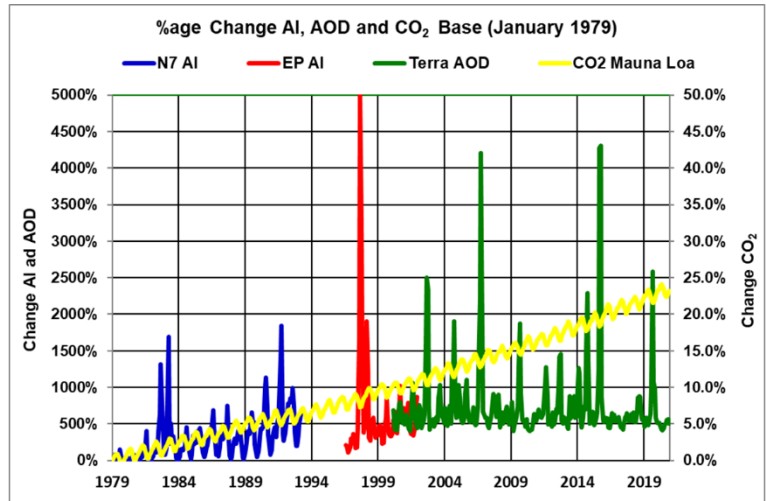

Figure 2: Percentage change in: average CSEAP Area, Aerosol Index (AI) (TOMS instrument - Nimbus 7 (N7); Earth Probe (EP); and AOD (Terra) with the Mauna Loa $CO_2$ (Keeling et al., 2005). Note: (1) the vertical scales differ by two orders of

magnitude; (2) the near random occurrence of significant peaks; and (3) the increasing trend in AI from 1979 to 2000.

The literature includes many attempts to connect volcanic eruptions and ENSO events. Nicholls (1988) and Nicholls (1990) investigated volcanic eruptions and El Niño events and concluded there was no connection. Hirono (1988) investigated the possibility that the 1983 eruption of El Chichon in Mexico could have triggered the El Niño which followed and Robock et

al. (1995) investigating this hypothesis found that: the eruption did not trigger the El Niño event which followed; and that only trade wind collapses in the western Pacific Ocean can initiate ENSO events. Handler and Andsager (1990) investigated the volcanic hypothesis which states that low-latitude volcanic aerosols are the immediate and only cause of warmer than normal SST or El Niño and its inverse using Monte Carlo techniques and found that both aspects of the hypothesis were satisfied. Self et al. (1997) investigated volcanic aerosol perturbations and the 16 strongest El Niño events over the last 150

140    years and found no general correlation. In discussing the evolution of ENSO events Trenberth et al. (2002) suggested that the effects of volcanic eruptions on ENSO events remained unanswered questions. Emile-Geay et al. (2008) focused on very





large eruptions which were greater than the Pinatubo eruption in 1991 and found that small eruptions have no effect. Zhang et al. (2013) investigated the effects of large eruptions using Aerosol Optical Depth (AOD) in latitudinal bands 0 to 30 and 30 to 90 in both hemispheres to force the climate model but did not consider the location or the intensity of the eruption and noted this should be done in the future. Cane (2005) reviewed forecasts of ENSO activity, found that there was no clear picture and suggested solar and volcanic variations in solar insolation and atmospheric aerosols might have a role. Ammann et al. (2003) found that including an improved volcanic eruption dataset in climate model simulations improved the correlation between the modelled data and observations whilst Mann et al. (2005) showed a short term response of ENSO to tropical eruptions. Other publications which address the link between volcanic eruptions and ENSO are (Timmreck, 2012), (Maher et al., 2015), (Blake et al., 2018) and (Predybaylo et al., 2017).

There is therefore obviously great interest in, but no general agreement on, the connection between volcanic eruptions and ENSO events for two reasons: In general, only large eruptions were considered and smaller eruptions which do not eject tephra into the stratosphere were ignored; and a global analysis was undertaken rather than focusing on a specific region, the SEAP Area, which is the only area which can create and sustain an ENSO event because of its location in the region which drives the entire non-ENSO Walker Circulation.

In this paper I report the connection of the volcanic tephra plumes emanating from within the SEAP Area and the ENSO indices.

### 1.2.3 ENSO return frequency

The literature describes ENSO events as exhibiting a return frequency of two to seven to ten years and the University Corporation for Atmospheric Research (UCAR) website at http://webext.cgd.ucar.edu/Multi-Case/CVDP_repository/cesm1.lm/nino34.powspec.png shows the ENSO power spectra for all the LME runs and the same information for the HadISST_1 (Rayner et al., 2003) and ERSST v5_1 (Huang et al., 2017). The HadISST_1 and ERSST v5_1 spectra are similar and Fig. 3(a) shows the multiple peaks in the HadISST_1 data whilst all the LME runs show a single peak at about 5 years e.g., Fig. 3(b) and are nearly identical even with the different forcings used and are all very different from the measured spectra. I suggest that the reason for the difference is that none of the forcings applied in the LME runs included volcanic forcing at the required spatial and temporal resolution and this is discussed later.

The combination of HadlSST_1 and SEAP Area volcanic tephra power spectra from 1980 to 2017 in Fig. 3(c) shows that the SEAP Area volcanic tephra power spectrum is similar in form to the HadlSST_1 spectrum with nearly identical multiple peaks including the major peaks at 6.17 and 5.42 years respectively. Given the uncertainties in the tephra data which is based on GVP Volcanic Explosivity Index (VEI) data available as integer values on a pseudo logarithmic scale and converted to tephra (VEIT) using Table 1 in (Newhall and Self, 1982) the similarity of the spectra is impressive.





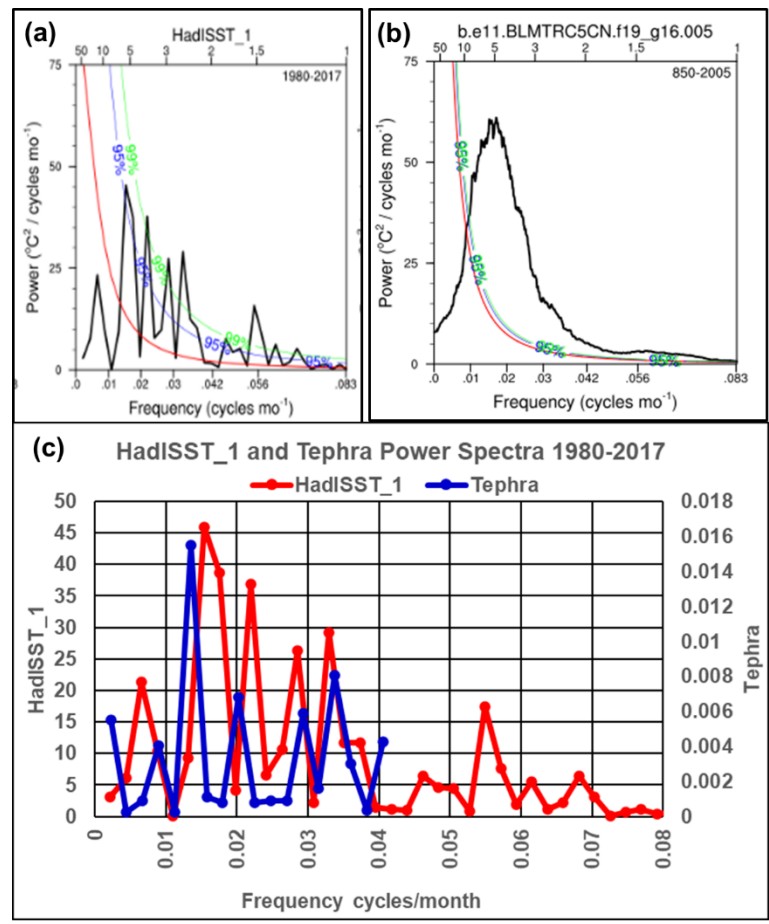

Figure 3: ENSO power spectra: HadISST_1 (a); LME all forcings run 5 (b); and combined HadlSST_1 and SEAP Area tephra (c).

## 1.2.4 ENSO seasonality

ENSO is highly seasonal as Fig. 4 shows and this paper provides an explanation for the seasonality. NCEP/NCAR average monthly rainfall in the CSEAP area and the Nino 3.4 SST (1980-2020) correlate at -0.57 significance <0.1 with the SST reducing when rainfall in the CSEAP Area is high. This clearly supports the hypothesis that the SEAP is the major cause of ENSO events as the SEAsian monsoon rainfall washes the aerosols out of the atmosphere in the SEAP Area enabling convection to be re-established in the region to drive the TWs and end the ENSO event.





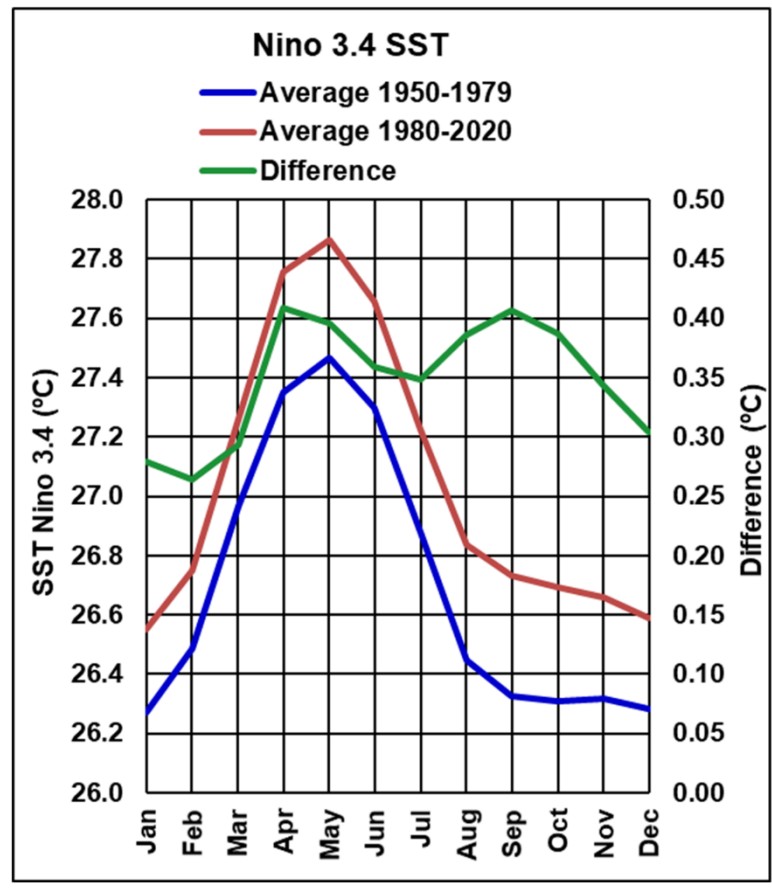

Figure 4: NCEP Reanalysis Nino 3.4 SST averages: 1950-79; 1980-2020; and the difference. Source NCEP Reanalysis

### 1.2.5 ENSO character -recent change

ENSO has become more intense in recent decades as Fig. 4 demonstrates with the Nino 3.4 SST rising by over 0.25°C in recent decades and especially in Aug-Nov when the anthropogenic SEAP is at its peak. This change in ENSO is noted in the IPCC AR6 Working Group 1 Draft Report at 2-97 (48-51) relying on Grothe et al. (2020) who state that in the recent five decades ENSO activity is 25% stronger than in the preindustrial era.

The general rise in all months is due to the increase in the levels of both the natural and anthropogenic SEAP as Appendices
A and B and Fig. 2 show and the anomalous rise from Aug to Nov is entirely due to the extreme apparitions of the anthropogenic SEAP.



## 1.3 Walker Circulation

The Walker Circulation is defined in the IPCC AR4 Glossary as a direct thermally driven zonal atmospheric circulation over the tropical Pacific Ocean, with convection in the west and sinking air in the east (See also (Wang, 2002), (Barry and Chorley, 2010), (Trenberth et al., 2000) and (Sturman and Tapper, 1996)).

The Australian Bureau of Meteorology (BOM) shows images of the atmospheric circulation during El Niño, La Nina and neutral seasons on its website at: http://www.bom.gov.au/climate/about/australian-climate-influences.shtml?bookmark=enso. It is clear from these figures that the "direct thermal drive" for the La Niña and neutral Walker Circulation must be located at ground level in the SEAP Area where solar radiation heats the Earth's surface which in turn heats the atmosphere as the rising limb of the non-ENSO Walker Circulation is located there in exactly the same location as the SEAP which is therefore uniquely positioned to directly affect the Walker Circulation and create ENSO events.

Since variations in the solar energy at the top of the atmosphere cannot explain the reduction in the surface heating in the SEAP Area which causes convection and the Walker Circulation to relax, the reduction must be caused by variations within the atmosphere where the SEAP reduces surface solar radiation. Indeed: Tosca et al. (2015) investigated the effects of anthropogenic aerosols in West Africa and reported that anthropogenic aerosols in the tropics limit convection; and Fig. 2.23 ((Nozawa et al., 2005) and (Takemura et al., 2005) in chapter 2 of the IPCC AR4 shows that, on a globally averaged basis, surface radiative forcing is controlled by aerosols with the net effect of long-lived greenhouse gases, ozone, aerosols and land use aligning nearly perfectly with the aerosol direct effect.

Therefore, from the IPCC AR4 and literature it is a plausible hypothesis that the SEAP could force the Walker Circulation and TWs to relax by reducing the surface solar radiation available to drive convection in the SEAP Area.

## 1.4 The Indian Ocean Dipole and ENSO

The IOD (Saji et al., 1999) is described on the Australian Bureau of Meteorology (BOM) website at http://www.bom.gov.au/climate/iod/ which shows images of the Indian Ocean atmospheric circulation in the three phases of the IOD and states that: IOD events start in May/June; peak a few months later and decay at the end of the southern spring when the monsoon arrives in the southern hemisphere. The BOM images clearly show that convection reverses over the SEAP Area when the IOD moves from a negative/neutral to positive phase.

Abram et al. (2020) found a persistent coupling has existed between the variability of the IOD and the El Niño/Southern Oscillation during the last millennium and the BOM website shows that the relaxing/reversing of convection in the SEAP Area is common to both. I show that the SEAP, uniquely positioned to control convection in the SEAP Area, drives both events simultaneously.



## 1.5    South Eastern Australia rainfall

Drought in SEAus has a major effect on agriculture and has been linked to ENSO and the IOD. The Centre for Australian Climate and Weather Research (CAWCR) produced technical report 26 (Timbal et al., 2010) which found the major agents correlating with drought are ENSO and higher MSLP in SEAus in the southern Autumn, Winter and Spring, whilst Ummenhofer et al. (2009) state that it is the IOD and not ENSO which drives drought in SEAus. I show that MSLP increases over SEAus from April to October, the wet season in SEAus, are driven by the SEAP perturbing the southern regional Hadley Circulation by forcing the associated tropical convection and sub-tropical high to the south.

Thus, I show the SEAP simultaneously creates ENSO and IOD events and drought in SEAus.

## 1.6    The South East Asian Plume (SEAP)

This paper focuses on SEAsia as this region is where convection, which drives the Walker Circulation, occurs and is therefore where aerosols can have a significant effect. Appendices 1 and 2 describe in detail the sources of aerosols in the SEAP Area, the natural SEAP derived from volcanic eruptions and the anthropogenic SEAP derived mainly from biomass burning and gas flares in the oil production industry. Reid et al. (2013) provide a review of the south east Asian aerosol system.

### 1.6.1    The natural SEAP

The SEAP Area, which covers about 3.4% of the globe, is the world's most tectonically active area with the United States Geological Survey (USGS) earthquake database (USGS, 2021) showing 25% (5 of 20) of the major earthquakes (magnitude > 8.4) in the world since 1900 occurred in the SEAP Area and the GVP database showing that from 1800 to 2020 over 26% of the global volcanic eruptions occurred in the SEAP Area. Simkin and Siebert (2000) state that 5 of 16 (31%) of the continuously erupting volcanoes in the world for the past 24 years are located in the SEAP Area and that one more, in Vanuatu, is just to the south east of the SEAP Area.

### 1.6.2    The anthropogenic SEAP

The anthropogenic SEAP is one of eight continental scale, anthropogenic, aerosol plumes which occur annually and are shown in Appendix B. These extreme plumes typically exist for a few months each year at the end of the regional dry season when biomass burning can occur. The SEAP is easily identified on the monthly mean AOD and AI data. The change in the CSEAP Area AI and AOD is shown in Fig. 2 and was extremely high in 1982, 1983, 1991, 1997, 2002, 2004, 2006, 2009, 2014, 2015 and 2019 compared with the intervening years. 1983 is the only year where an extreme peak emission is not in September or October. The Terra AOD data is shown in greater detail in Appendix B to demonstrate clearly the peak anthropogenic aerosol emission season is SON, the end of the dry season in SEAsia,





CALIPSO (Cloud-Aerosol Lidar and Infrared Pathfinder Satellite Observations) (Winker et al., 2009) profiles confirm a layer of smoke existed at about 3Km altitude in October 2015 in the SEAP Area and Fig. 5 shows the geographic extent of the 2015 extreme apparition of the SEAP. This paper analyses the effects of the SEAP on an annual and SON (when the anthropogenic plume is most intense and therefore has its greatest effect) basis.

The maximum SEAP AOD was 1.60 (Oct 2015) and the maximum Nimbus 7 (N7) and Earth Probe (EP) AI was 1.81 (Sept

1997). The AI of the CSEAP Area increased by about 500% from Jan 1979 to 1992 and by slightly more in 2000 in years without extensive biomass burning. From Jan 1979 to Sep 1997, a major biomass burning event year (Applegate et al., 2001), the increase in September AI was 5,000%. Note the AI and AOD in 2000/01 are the same magnitude and the AOD percentage increase has been calculated on this basis.

The surface radiative forcing of the anthropogenic SEAP is significant and the literature includes:

A 10% to 30% reduction of Photosynthetically Active Radiation (Ramanathan, 2006);

-150 W m-2 (Duncan et al., 2003); and during ACE-Asia, ~-286.0 W/m$^2$/τα (Hansell et al., 2003).

Figure 7 in Ott et al. (2010) shows the atmospheric heating effect of the SEAP in 2006 with temperature rises of over 0.5° C in October from 400 to 200 hPa.

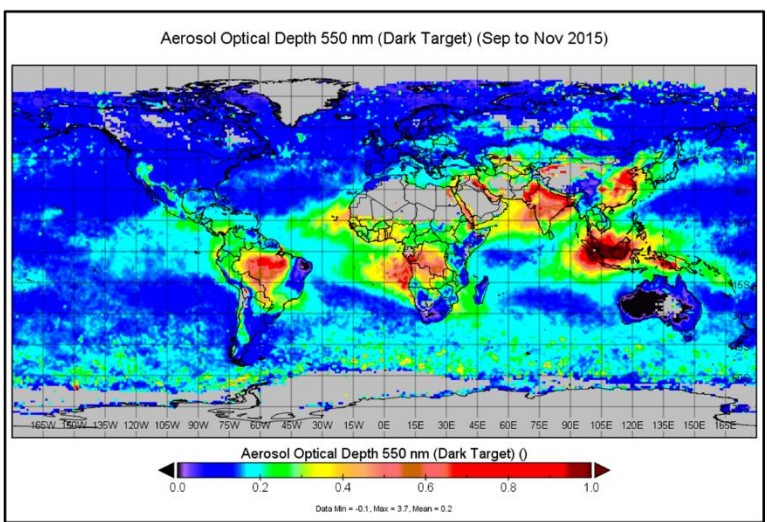

Figure 5: NASA Terra/MODIS AOD in SON 2015. Source: NASA Giovanni.

## 2  METHOD AND DATA

The creation of ENSO events by the natural and anthropogenic SEAP is demonstrated by showing that each step in the development of an ENSO event outlined in the introduction is directly linked to the SEAP in three ways:

**First**: the GVP data is used to demonstrate a direct link from the tephra ejected by volcanoes in the SEAP Area to the ENSO

indices and other associated events.



**Second**: modelling, reanalysis and satellite data is used to show the same effects as tephra and that ENSO and associated events can all be explained by physical mechanisms controlled by the SEAP.

**Third**: The SON data from 2015 and 2016 are used to show the effects of an extreme apparition of the SEAP.

### 2.1 Volcano data

Volcanic eruption tephra was derived from the GVP database using only confirmed eruptions, calculated as shown in Appendix C and analysed using the segments in Table 1. The other parameters used were: the NOAA ENSO indices (Nino 3.4 and 1+2 SST); BOM SOI; NCEP reanalysis SST, omega and wind speed; BOM 9:00am sea level pressure and rainfall for south eastern Australia; and the NOAA Dipole Mode Index (IOD) and these datasets limited the time span of the analysis to the periods in Table 1.

This data was not detrended due to the massive eruptions of Krakatau (1883) and Agung (1963) distorting the detrending process

| | | GVP Data Segments (Km$^3$) | Period |
|---|---|---|---|
| 1 | Surface Temp SEAP Area | Standard | 1948-2020 |
| 2 | Omega SEAP Area | Standard | 1948-2020 |
| 3 | Wind Speed Nino 3.4 Area | Standard | 1948-2020 |
| 4 | Nino 3.4 SST | Standard | 1870-2020 |
| 5 | Nino 1+2 SST | 0, 0.00054, 0.001, 0.004, >0.004 | 1870- 2020 |
| 6 | SOI | Standard | 1876-2020 |
| 7 | Convection Nino 3.4 Area | 0, 0.007, 0.02, > 0.02 | 1948-2020 |
| 8 | IOD (May to October) | Standard | 1870-2020 |
| 9 | Pressure SE Australia (Apr-Oct) | 0.00035, 0.0004,0.001, 0.01, >0.01 | 1903-2020 |
| 10 | Rainfall SE Australia (Apr-Oct) | 0, 0.003, 0.005, 0.001, 0.005, 0.1, > 0.1 | 1870-2020 |

Table 1: GVP volcano data segments and analysis periods. Standard segment boundaries are 0, 0.0001, 0.001, 0.01, 0.1 and 285 greater than 0.1 Km$^3$. Periods other than 1870-2020 are limited by the availability of BOM data except for 1948-2020 which is limited by the NCEP/NCAR reanalysis data.

### 2.2 Modelling, reanalysis and satellite data

All this data was detrended by: copying all the individual time series (88 for the LME, 11 for MERRA-2 (hereafter M2) and 22 for TN (annual + SON)) to PAST 3 (Hammer et al., 2001), a statistical analysis software package; detrending the data (a 290 simple one-click process); copying the detrended data back to a spreadsheet; and adding the average of the un-detrended data to each element of the time series to give a series with the same average magnitude as the un-detrended data which enables a direct comparison between measured data and the LME, M2 and TN data.





### 2.2.1 Last Millennium Ensemble (LME)

The LME (Otto-Bliesner et al., 2016) data analysed is: atmosphere, post processed monthly averages consisting of seven
simulations (Table 1 (Otto-Bliesner et al., 2016)) plus "850" which were repeated in multiple runs. Aerosol Optical Depth
(AOD) (AODVIS), surface pressure (PSL), surface temperature (TS), precipitation (PRECL), vertical pressure velocity
(OMEGA) and wind speed (U10) data from these simulations(run) was downloaded: (850 (3); All (13); Ozone and Aerosol
(2); Green House Gas (3); Land Use (3); Orbital (3); Solar (5); and Volcanic (5). Each simulation(run) provided two
NetCDF data files for each parameter covering the entire world from 850 to 1849 and 1850 to 2005 except for the ozone and
aerosol simulation which provided one file from 1850 to 2005. Matlab (2019) was then used to: extract data for the required
areas from each file; create a single time series of 1,156 years (13,872 months); and export the time series data and date
together with the latitudes and longitudes used to a spreadsheet.

The time series data was then converted to tabular form by year and month, checked for accuracy by summation of the time
series and table, and the average (total for rainfall) annual data calculated. This data was then detrended as described above.
Finally, data for each parameter from all eight simulations/runs was plotted on the same scatter plot against the AOD for the
same simulation/run to give 8,248 points, the trend and $R^2$ value.

Climate indices for the IOD and SOI are not available directly from the LME. The IOD was calculated using TS from the
standard IOD areas. The SOI was calculated using the BOM formula at: http://www.bom.gov.au/climate/enso/soi/about-
soi.html and PSL data for the areas near Darwin (12º to 14º S and 130º to 133º E) and Tahiti (16º to 18º S and 207º to 213º
E).

**Note:**

(1)     TS (skin temperature) from the atmosphere dataset was used rather than SST (sea surface temperature) from the
ocean dataset for simplicity as the ocean dataset is provided on a rotated grid rather than latitude and longitude.

(2)     Although the annual LME SEAP Area AODVIS data ranges from 0.04 to 0.11 which is significantly lower than the
Terra annual data which ranges from 0.18 to 0.27, the data still shows statistically significant relationships between
the LME SEAP Area AOD, the ENSO indices and the other parameters.

(3)     The volcanic forcing data in the LME cannot of itself demonstrate the connection between the natural volcanic SEAP
and ENSO as it does not have the resolution required to do so. It is derived from ice cores in the Arctic and Antarctic
(Otto-Bliesner et al., 2016) and Gao et al. (2008) who provide the data as only stratospheric sulphate forcing in
latitude bands (ten degrees wide), by altitude and month. For volcanic tephra to travel to the polar regions the tephra
must be injected into the stratosphere requiring a minimum VEI of 3 to 4. Hence all the VEI 0,1,2 and some (50%
assumed) of the VEI 3 eruptions in the SEAP Area must be missing from the dataset. Since this excludes over 98% of
the eruptions in the SEAP Area since 1870 which are used in this paper the resolution of this LME volcanic forcing
dataset is inadequate in itself in terms of eruption size, geo-space, time and aerosol type to prove the causation of
ENSO by SEAP Area volcanic tephra described in this paper. However, the LME volcanic forcing does provide



another independent aerosol forcing dataset which shows the same results as the other LME simulations and is therefore included in the analysis.

### 2.2.2 MERRA-2 reanalysis

A similar approach was used for the NASA M2 reanalysis dataset (1980-2020) which includes assimilated aerosols. AOD
(M2IMNXGAS_5_12_4_AODANA), surface pressure (M2IMNPANA_5_12_4_PS) precipitation (M2TMNXFLX_5_12_4_PRECTOT) surface skin temperature (M2TMNXSLV_5_12_4_TS), omega (M2IMNPASM_5_12_4_OMEGA) and surface eastward wind (mean_M2TMNXFLX_5_12_4_ULML) data was downloaded, and annual averages or totals calculated. All M2 data was detrended in the same way as the LME data.
The SOI and IOD for the M2 analysis were calculated using data from the M2 dataset using the BOM formula for the SOI
and surface pressure near Darwin (130º to 131º E and 12º to 13º S) and Tahiti (149º to 150º W and 17º to 18º S) and TS for the standard IOD areas

### 2.2.3 Satellite data

AOD data from the Terra satellite (2000-2020) for the SEAP Area was downloaded together with the NCEP reanalysis sea surface temperature (SST), omega, sea level pressure, precipitation rate and zonal wind. Annual and SON time series were
calculated for all these parameters and the NOAA Nino 3.4 and 1+2 SST, the BOM SOI and the NOAA IOD. The SON data is used to show the significant effect the extreme anthropogenic SEAP has in SON.

### 2.3 Data sources

| | |
|---|---|
| LME: | https://www.earthsystemgrid.org/dataset/ucar.cgd.ccsm4.CESM_CAM5_LME.html |
| MERRA-2: | https://giovanni.gsfc.nasa.gov/giovanni/ |
| GVP | https://volcano.si.edu/list_volcano_holocene.cfm |
| Terra | https://giovanni.gsfc.nasa.gov/giovanni/ |
| NCEP: | https://psl.noaa.gov/cgi-bin/data/timeseries/timeseries1.pl |
| SST | Niño 3.4 area (5º S-5º N, 170º W-120º W) and Nino 1+2 areas (10°S–0°, 90°W–80°W). |
| | http://www.cpc.ncep.noaa.gov/data/indices/ |
| SOI | http://www.bom.gov.au/climate/current/soihtm1.shtml |
| ENSO Indices | https://psl.noaa.gov/gcos_wgsp/Timeseries/ |
| IOD | https://psl.noaa.gov/gcos_wgsp/Timeseries/DMI/ |
| Omega | (vertical motion in the atmosphere) at 400mb level - CSEAP Area. http://www.esrl.noaa.gov/psd/cgi-bin/data/timeseries/timeseries1.pl (Source 1) |





SST SEAP Area     (Source 1)

TW Index (TWI) (850mb at 5º N-5º S, 175º W-140º W) http://www.cpc.ncep.noaa.gov/data/indices/cpac850 (Source 2)

Pressure and rainfall SEAus:      BOM climate data cd to 2010 and Source 1

## 3    RESULTS

Results: are shown in the order of the sequence of events required to create an ENSO event outlined in the introduction; the
specific stage is noted in the title; and are presented as:

1. Scatter plots of the segmented volcano data with the volcanic tephra shown on a logarithmic scale. The segment boundaries used are listed in Table 1;

2. Scatter plots of the LME, M2 and TN annual and SON data which demonstrate the effects of the very high SEAP Area AOD in SON;

3. Plots of the difference between the 2015 (high) and 2016 (low) AOD SON seasons which indicate visually the changes wrought by this extreme event using NCEP/NCAR reanalysis and M2 data via NASA Giovanni and Panoply;

4. A listing of the correlations shown on the graphs in Table 2 with: change per unit AOD; change over tephra/AOD range; and percentage change in rainfall.

### 3.1    Stage 1: The SEAP forms

The 2015 SON SEAP AOD was at extreme levels and Fig. 6 shows the 2015/2016 comparison with Borneo and most of Sumatra, the CSEAP Area, covered by AOD increases over 1.0 and over half the SEAP Area showing increases over 0.2.



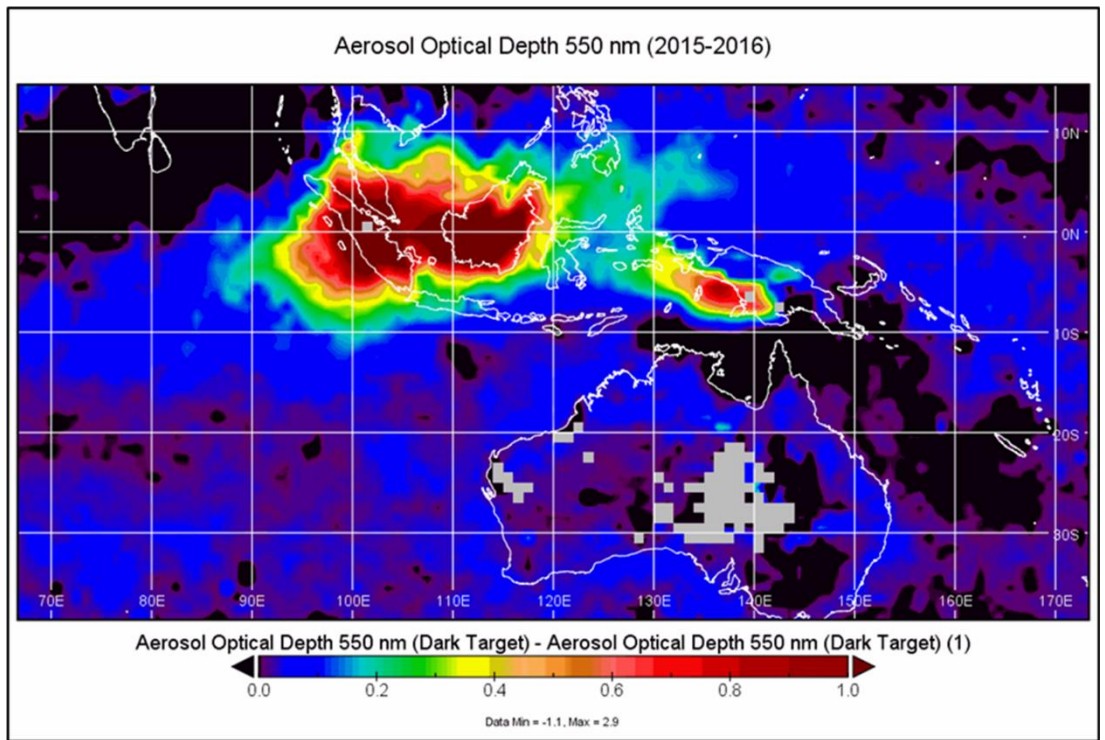

Figure 6: Terra/MODIS AOD SON 2015-2016 (Source NASA Giovanni and Panoply)

## 3.2    Stage 2(a): The SEAP warms the troposphere

Ott et al. (2010) investigated the extreme SEAP of Aug-Nov 2006 and Figures 6, 7 and 8 in the paper clearly show atmospheric heating due to the aerosols from 1,000 to 150 hPa.

### 3.3    Stage 2(b): The SEAP reduces surface solar radiation cooling the surface

Investigating the effects of the extreme 1997 biomass burning in Indonesia in SON Rajeev et al. (2008) found that this event caused surface radiative forcing of over $-46 W/m^2$ and more than 1° C cooling of the surface. Table 1 shows that standard segments are used for the volcano data and Table 2 shows that increasing levels of tephra and aerosols result in lower surface temperatures in the SEAP Area. Figure 7 (c) shows a reduced SST of over 1.5 ° K in the SEAP Area south of the equator.





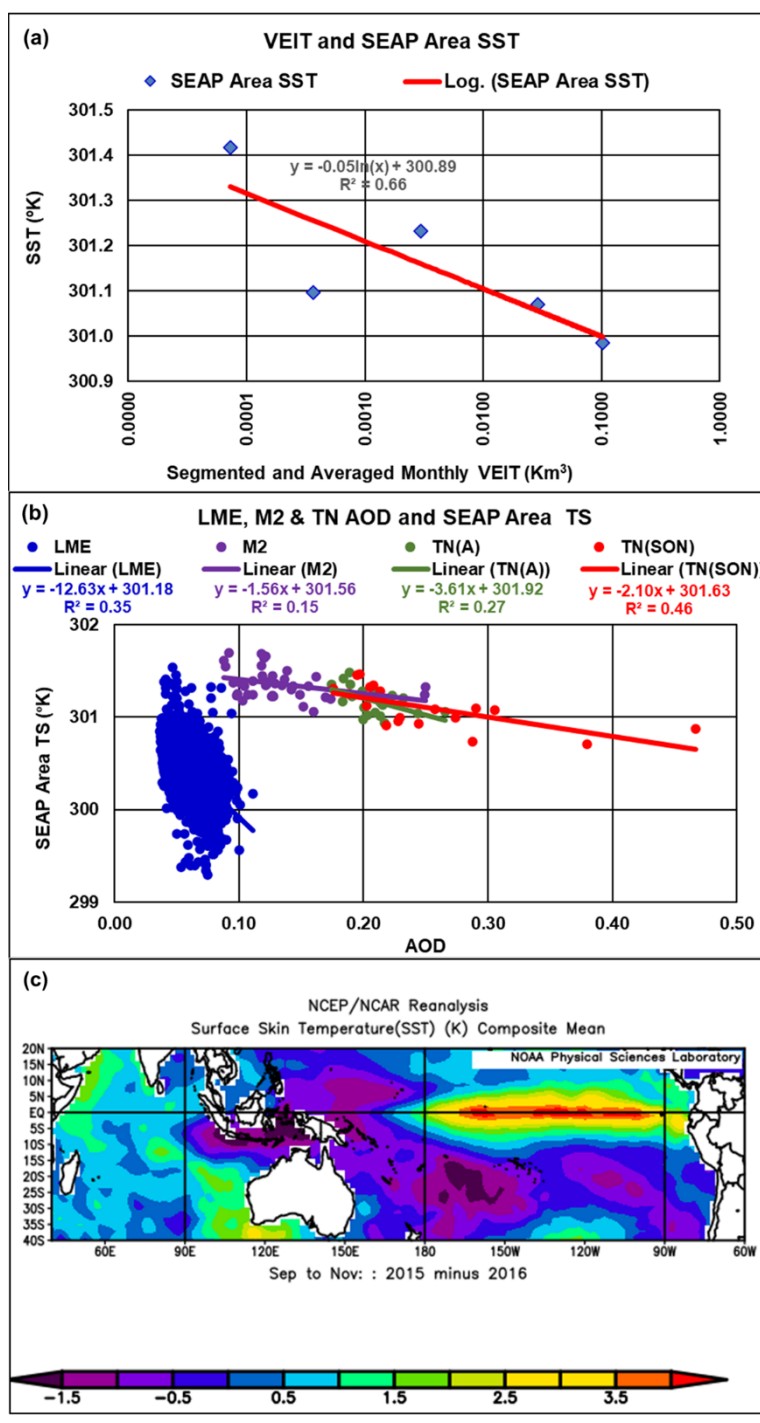

Figure 7: Scatter plots showing: segmented annual SEAP Area VEIT and SST (a); and LME, M2 and TN annual and TN
SON SEAP Area AOD and TS (b). NCEP/NCAR SON SST 2015 minus 2016 (c)





### 3.4 Stage 3: Reduced SEAP Area SST reduces convection

Table 1 shows that standard segments are used for the volcano data and Table 2 shows that higher levels of tephra and aerosols result in increased omega and thus reduced convection Figure 8(c) shows omega increasing by over 0.04 Pa/s over most of the SEAP Area.

**Note:** Omega is vertical pressure velocity in the atmosphere. Positive values indicate falling motion and negative values rising, and increased omega therefore means reduced convection.






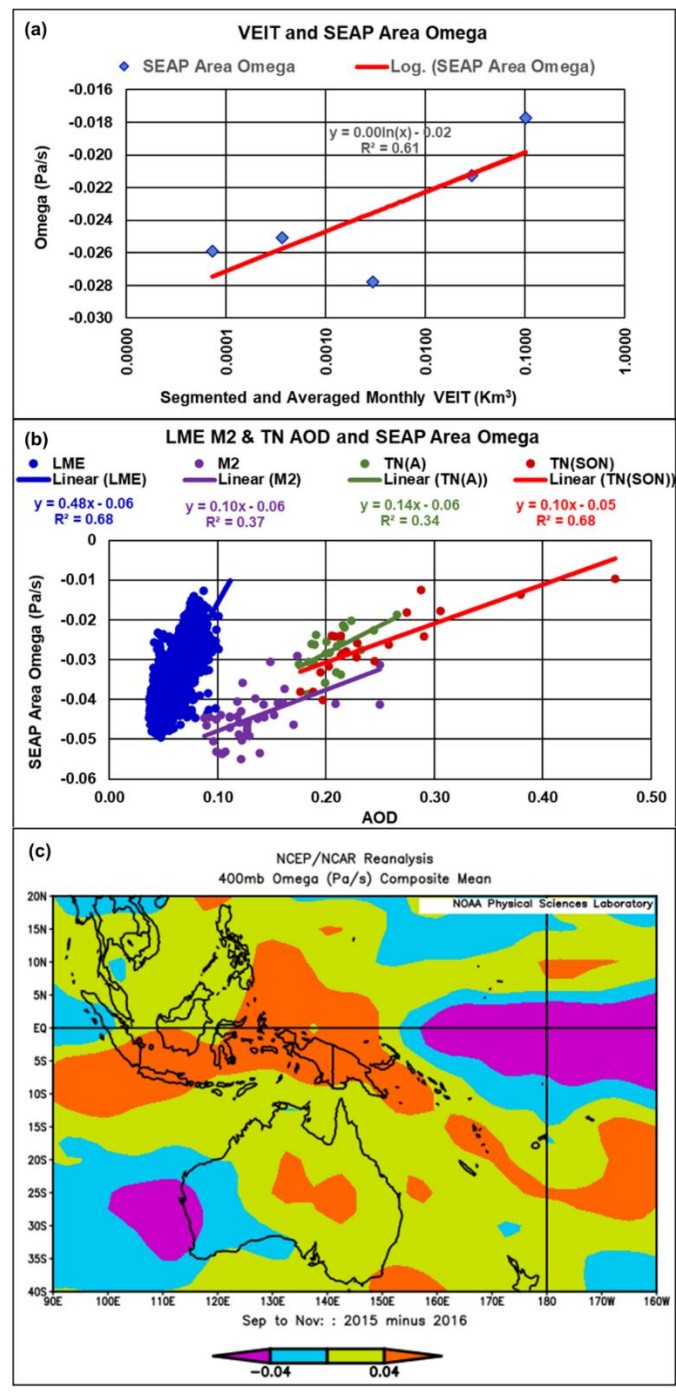

Figure 8: Scatter plots showing: segmented annual SEAP Area VEIT and omega (a); and LME, M2 and TN annual and TN SON SEAP Area AOD and omega (b). NCEP/NCAR SON omega 2015 minus 2016 (c)





### 3.5 Stage 4: Reduced convection in the SEAP Area causes the trade winds to relax

Table 1 shows the standard segments used for the volcano data and Table 2 shows that increasing levels of tephra and
aerosols result in reducing TW velocity. Figure 9 (c) shows the changes in the surface vector winds and it is clear that the
TWs which normally blow east to west across the Pacific Ocean at about 6m/s have effectively reversed near the equator and
the dateline in 2015 as an inspection of the data in the individual years which make up Fig. 9 (c) confirms.

**Note**: wind direction has been aligned across the datasets and positive velocity denotes winds blowing from east to west.






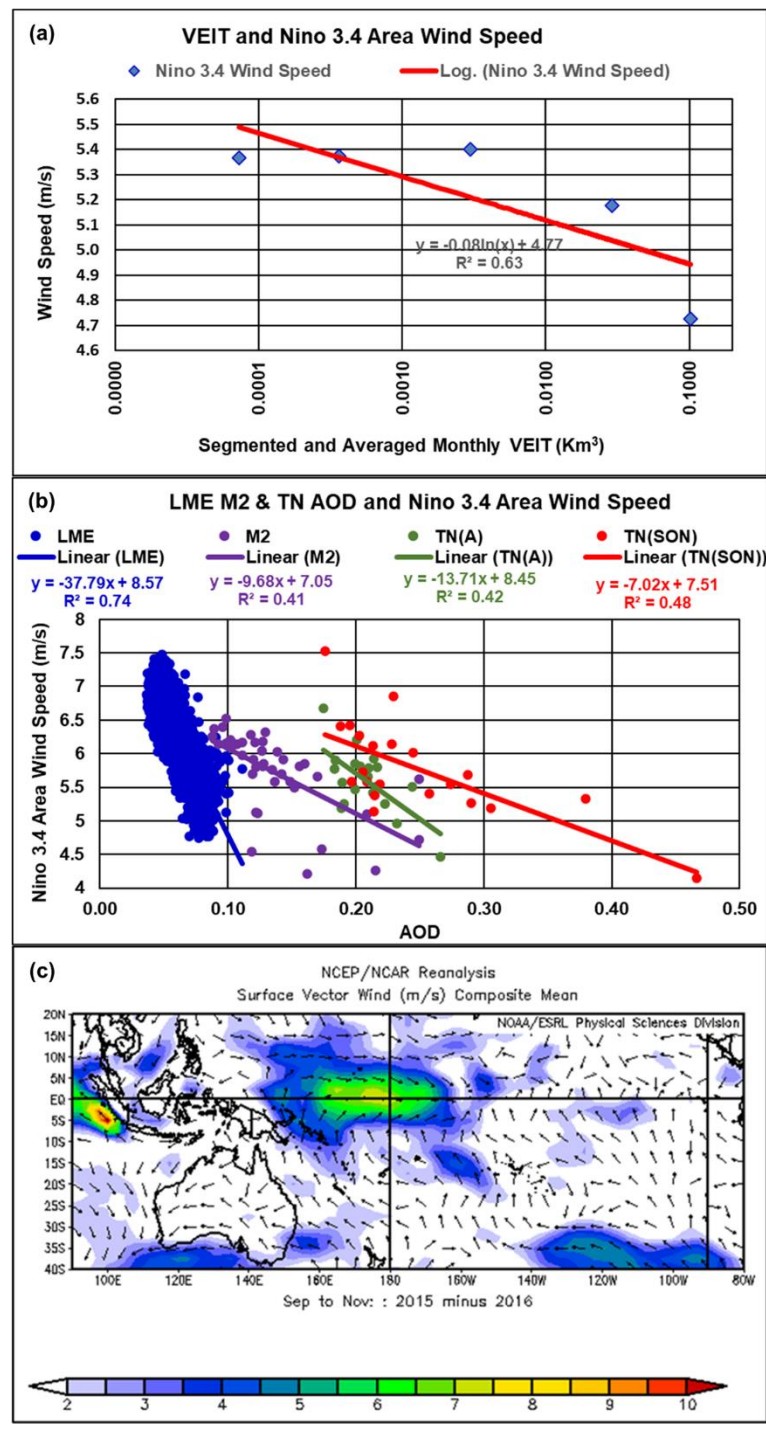

Figure 9: Scatter plots showing: segmented annual SEAP Area VEIT and Nino 3.4 area wind speed (a); and LME, M2 and TN annual and TN SON AOD and Nino 3.4 wind speed (b). NCEP/NCAR SON wind speed 2015 minus 2016 (c)





### 3.6 Stage 5: Wind Speed and Nino 3.4 Area TS

Table 2 shows that reduced TW speed in the Nino 3.4 area results in increased Nino 3.4 SST.

Figure 10 shows that, consistent with the literature, TS in the Nino 3.4 area is controlled by wind speed with reduced wind speed forcing annual average TS changes of up to 4.88° K (LME) which is consistent with the creation of an ENSO event

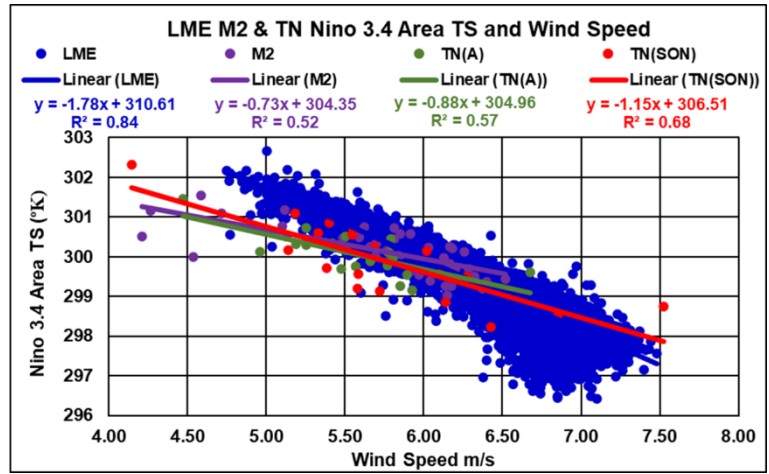

Figure 10: Scatter plot showing LME, M2 and TN annual and TN SON Nino 3.4 area TS and wind speed.

### 3.7 Niño Areas SST and the SOI

These are the prime indices used to monitor the onset and progress of an ENSO event.






### 3.7.1 Stage 5: Nino 3.4 SST

Table 1 shows the standard segments used for the volcano data and Table 2 shows that increased levels of tephra and aerosols result in increased Nino 3.4 SST. Figure 7 (c) shows a classic warm tongue extending from South America across

the Pacific Ocean with the Nino 3.4 SON SST rising between 2 and 4+ °K which clearly indicates an ENSO event is occurring.

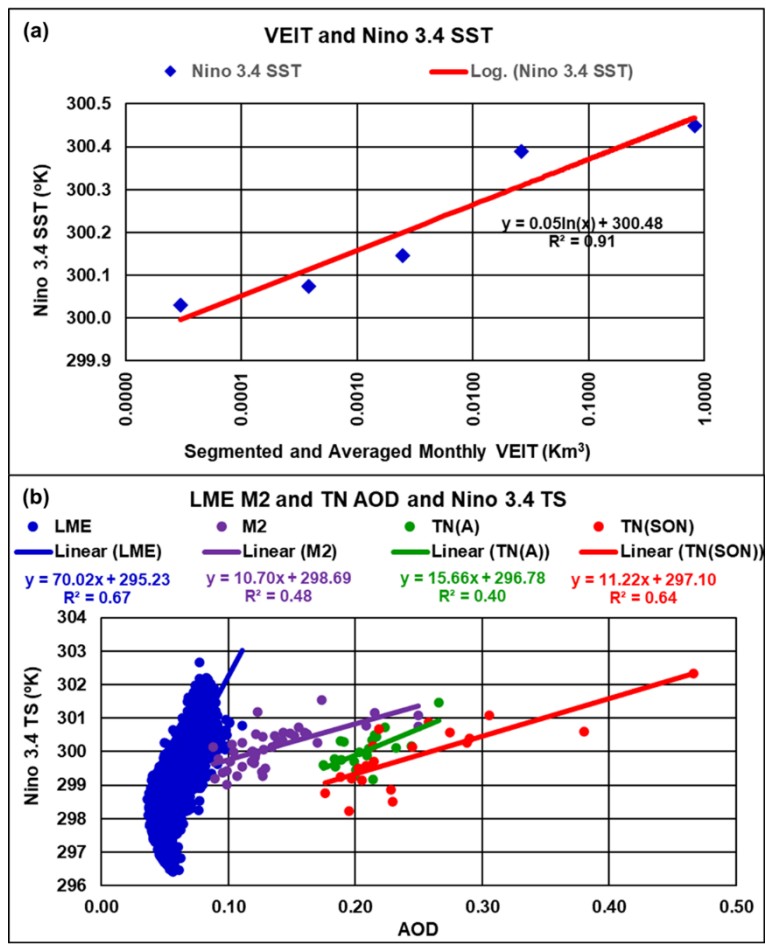

Figure 11: Scatter plots showing: segmented annual SEAP Area VEIT and Nino 3.4 SST (a); and LME, M2 and TN annual

and TN SON TS and AOD (b). Figure 7 (c) shows the NCEP/NCAR 2015 minus 2016 SON SST.





### 3.7.2    Stage 5: Nino 1+2 SST

Table 1 shows the segments used for the volcano data and Table 2 shows that increasing levels of tephra and aerosols result

in increased Nino 1+2 SST. Figure 7 (c) shows the Nino 1+2 SON SST rising by between 0.5 and 3.5 °K

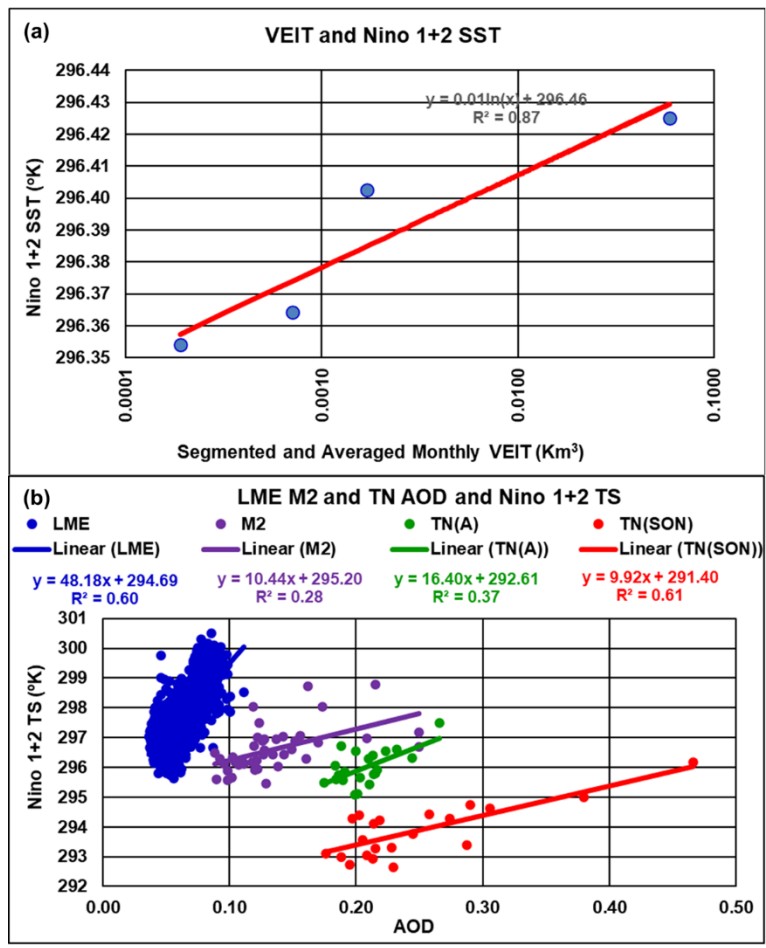

Figure 12: Scatter plots showing: segmented annual SEAP Area VEIT and Nino 1+2 SST (a); and LME, M2 and TN annual and TN SON Nino 1+2 TS and AOD (b). Figure 7 (c) shows the NCEP/NCAR 2015 minus 2016 SON SST.


### 3.8    Stage 6: Higher Nino 3.4 SST causes convection

Cai et al. (2015) investigating the potential changes in ENSO due to greenhouse warming note that atmospheric convection follows the highest sea surface temperature and this is confirmed here.

Table 1 shows the segments used for the volcano data and Table 2 shows that increasing levels of tephra and aerosols result

in reducing omega in the Nino 3.4 area and thus increasing convection. Figure 13(c) shows a vertical cross section of omega





averaged across the SEAP Area latitudes with values reducing by 0.03 Pa/s in the central Pacific Ocean increasing by over 0.02 Pa/s over the SEAP Area which forces the Walker Circulation to relax/reverse.

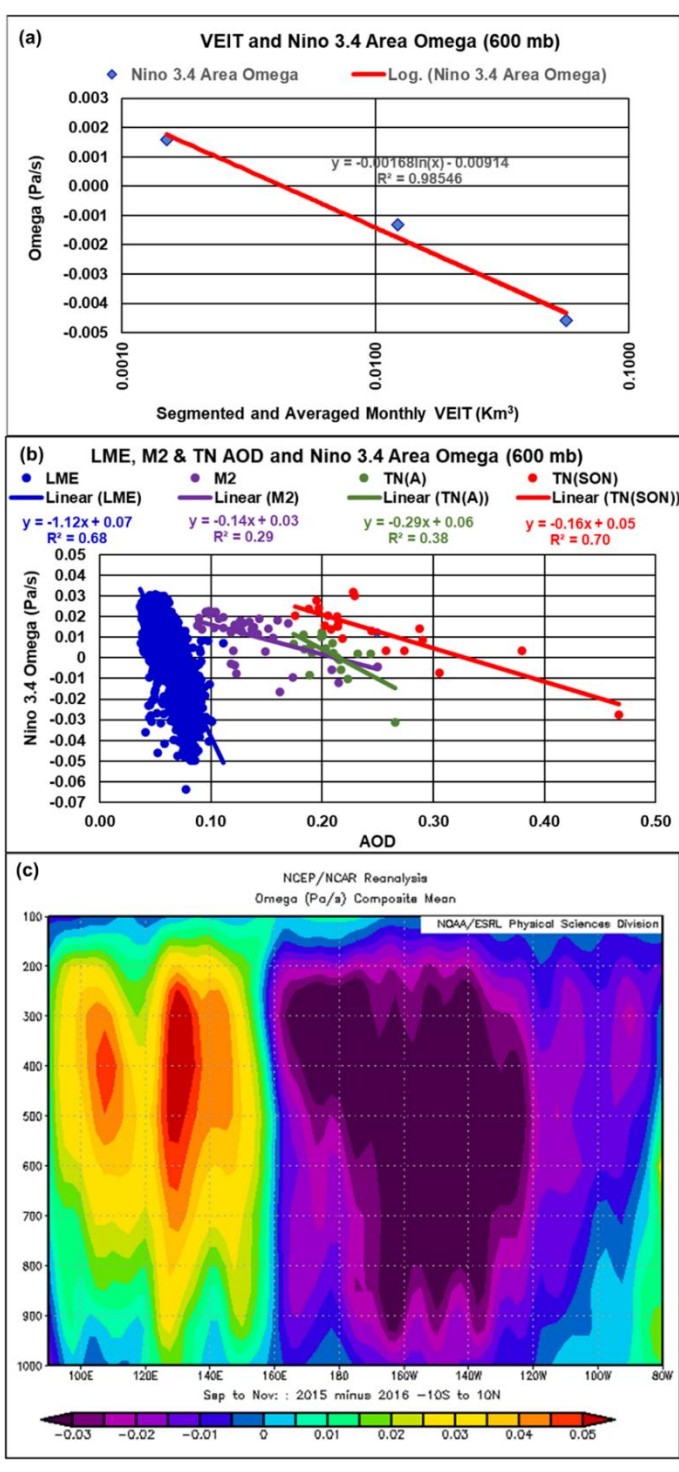



Figure 13: Scatter plots showing: segmented annual SEAP Area VEIT and Nino 3.4 area omega (a); and LME, M2 and TN
annual and TN SON SEAP Area AOD and Nino 3.4 Area omega (b). NCEP/NCAR SON omega by longitude and height
averaged across the SEAP Area latitudes 2015 minus 2016 (c).






### 3.8.1    Stage 7: Southern Oscillation Index

Table 1 shows the standard segments used for the volcano data. The data is limited to 1876-2020 due to the availability of BOM SOI Data and Table 2 shows that increasing levels of tephra and aerosols result in lower values of the SOI. Figure 14(c) shows increased pressure in Darwin and reduced pressure in Tahiti which drove the SOI negative in 2015. The
pressure changes are created by increased SST/convection in the central Pacific Ocean which reduces surface pressure and reduced SST/convection in the SEAP Area which raises surface pressure.



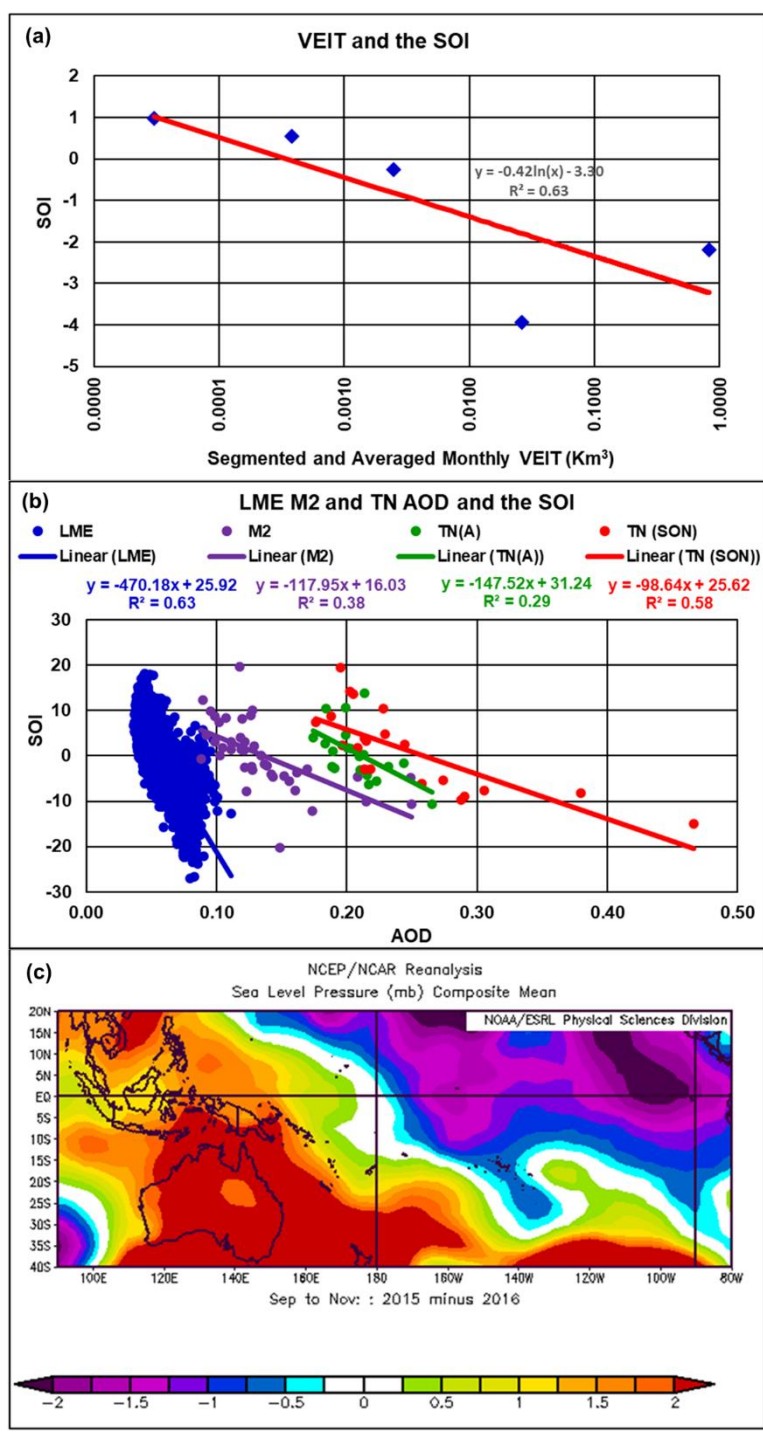

Figure 14: Scatter plots showing: segmented annual SEAP Area VEIT and the SOI (a); and LME, M2 and TN annual and TN SON SEAP Area AOD and the SOI (b). NCEP/NCAR SON sea level pressure 2015 minus 2016 (c).



### 3.9 Other effects commonly associated with ENSO

Simultaneously with creating ENSO events the SEAP also creates Indian Ocean dipole events and drought in SEAus.

### 3.9.1 The Indian Ocean Dipole (IOD) – May to October

Table 1 shows the standard segments used for the volcano data and Table 2 shows that increasing levels of tephra and
aerosols result in a positive change in the IOD. Figure 7(c) shows most of the IOD east area is cooler and most of the west area is warmer.

**Note:** The SEAP covers the eastern area used to calculate the IOD and therefore has a direct and immediate impact on one element of the IOD calculation

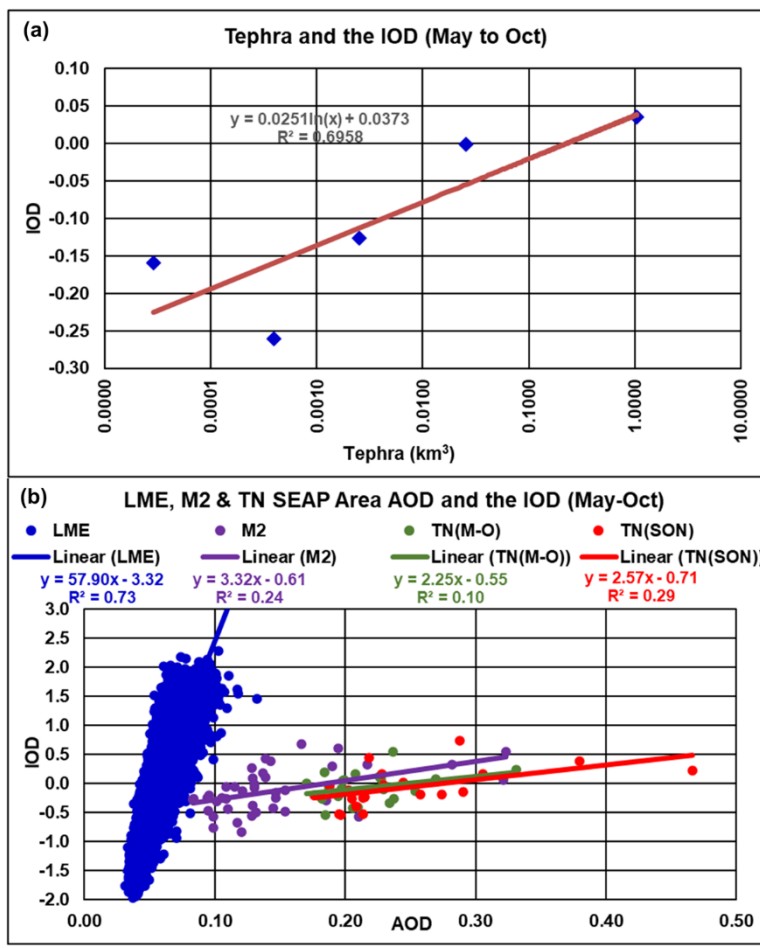


Figure 15: Scatter plots showing: segmented annual SEAP Area VEIT and the IOD (a); and LME, M2 and TN annual and TN SON SEAP Area AOD and the IOD (b).





### 3.9.2    Pressure south eastern Australia (April to October)

As well as perturbing the Walker Circulation by reducing convection in the SEAP Area, the SEAP simultaneously moves convection in the southern regional Hadley circulation south which moves the regional sub-tropical high south creating anomalous high pressure over SEAus.

Table 1 shows the segments used for the volcano data for the period 1903-2020 which is limited by the availability of BOM data. Pressure data is from the BOM station 86071 at 9:00 am 1903 to 2008 and NCEP/NCAR 2009 to 2020 for 144° to 146° E and 37° to 38° S. The two data sets correlate well between 1948 and 2008 at 0.96.

Table 2 shows that increasing levels of tephra and aerosols result in increased pressure over SEAus and Figure 16 (c) shows the perturbation of the southern Hadley circulation in 2015 compared to 2016 with convection reduced over the SEAP Area, increased over the central and northern Australian continent and reduced over SEAus – consistent with the regional Hadley Circulation moving south. Figure 14 (c) shows the SON sea level pressure change 2015 minus 2016.


Atmospheric Chemistry and Physics Discussions — Open Access

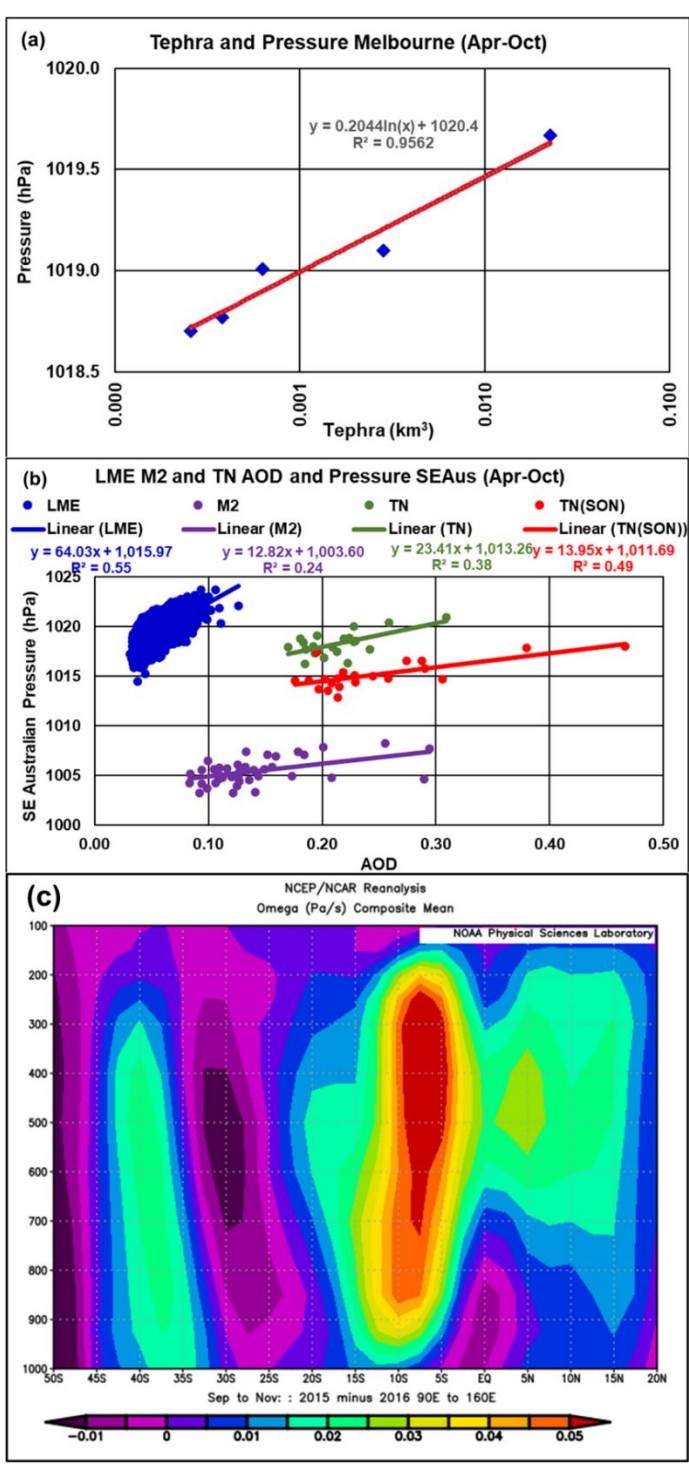





Figure 16: Scatter plots showing: segmented annual SEAP Area VEIT and pressure SEAus (a); LME, M2 and TN Apr-Oct and TN SON AOD and pressure SEAus (b). NCEP/NCAR SON omega by latitude and height averaged across the SEAP

Area longitudes 2015 minus 2016 (c).

### 3.9.3    Rainfall south eastern Australia (April to October)

Table 1 shows the segments used for the volcano data. Rainfall data is from the BOM Melbourne station 86071 (1870-2014) and station 86039 (2015-2020).        Average rainfall for the month was inserted into the three months without data.

Table 2 shows that increasing levels of tephra and aerosols result in reduced rainfall in SEAus.

Figure 17(c) shows rainfall reducing by over 1 mm per day in SEAus in SON. which, given that the BOM rainfall data from 1870 to 2020 shows the average daily rainfall in Melbourne in SEAus in SON is only 1.7 mm/day, implies that the SEAP reduced rainfall in coastal SEAus by over 50% and by a greater amount further inland in the grain growing region near Swan Hill where the total average rainfall in SON is about 1.0 mm/day.

This finding is of exceptional importance for SEAus grain growers who lost over A\$2 billion in revenue in 2006 because

crops failed due to drought in a year when volcanic activity was at high levels and the SON AOD of the SEAP was extreme. It is therefore a perfect fit for this special issue of Climate of the Past - "Volcanic Impacts on Climate and Society".

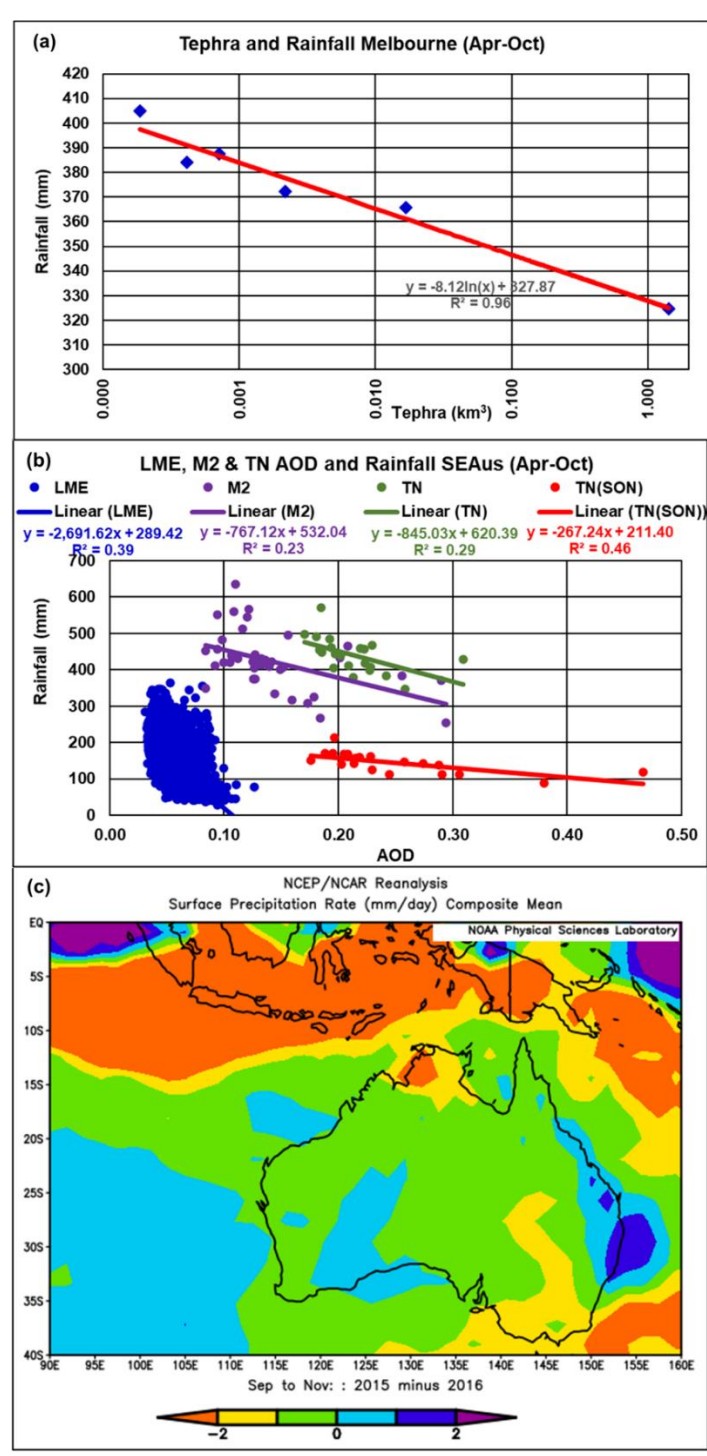

Figure 17: Scatter plots showing: segmented April to October SEAP Area VEIT and rainfall Melbourne (a); and LME, M2 and TN April to October and TN SON AOD and rainfall SEAus (b). NCEP/NCAR SON surface precipitation 2015-2016 (c)



| A | Surface Temp SEAP Area | GVP | LME | MERRA-2 | TN (SON) |
|---|---|---|---|---|---|
| 1 | R | -0.81 | **-0.59** | *-0.38* | **-0.68** |
| 2 | $R^2$ | 0.66 | **0.35** | *0.15* | **0.46** |
| 3 | Trend/unit AOD (° K) | | -12.63 | -1.58 | -2.10 |
| 4 | Change AOD/tephra range (° K) | -0.33 | -1.06 | -0.25 | -0.61 |
| B | **Omega SEAP Area** | | | | |
| 1 | R | 0.78 | **0.82** | **0.61** | **0.83** |
| 2 | $R^2$ | 0.61 | **0.68** | **0.37** | **0.68** |
| 3 | Trend/unit AOD (Pa/s) | | 0.48 | 0.10 | 0.10 |
| 4 | Change AOD/tephra range Pa/s | 0.008 | 0.035 | 0.02 | 0.03 |
| C | **Wind Speed Nino 3.4 Area** | | | | |
| 1 | R | -0.79 | **-0.86** | **-0.64** | **-0.69** |
| 2 | $R^2$ | 0.63 | **0.74** | **0.41** | **0.48** |
| 3 | Trend/unit AOD (m/s) | | **-37.8** | **-9.68** | **-7.02** |
| 4 | Change over AOD range (m/s) | -0.55 | **-2.76** | **-1.56** | **-2.04** |
| D | **Wind Speed and TS Nino 3.4 Area** | | | | |
| 1 | R | | **-0.92** | **-0.72** | **-0.83** |
| 2 | $R^2$ | | **0.84** | **0.52** | **0.68** |
| 3 | TS/unit TW speed change | | **-1.78** | **-0.73** | **-1.15** |
| 4 | Change over AOD range | | **-4.88** | **-1.69** | **-3.88** |
| E | **Nino 3.4 SST** | | | | |
| 1 | R | 0.95 | **0.82** | **0.69** | **0.80** |
| 2 | $R^2$ | 0.91 | **0.67** | **0.48** | **0.64** |
| 3 | Trend/unit AOD (° K) | | **70.02** | **10.70** | **11.22** |
| 4 | Change over AOD range (° K) | 0.47 | **5.11** | **1.72** | **3.27** |
| F | **Nino 1+2 SST** | | | | |
| 1 | R | 0.93 | **0.78** | **0.53** | **0.78** |
| 2 | $R^2$ | 0.87 | **0.60** | **0.28** | **0.61** |
| 3 | Trend/unit AOD (° K) | | **48.18** | **10.4** | **9.92** |
| 4 | Change over AOD range (° K) | 0.07 | **3.52** | **1.68** | **2.89** |
| G | **SOI** | | | | |
| 1 | R | -0.79 | **-0.80** | **-0.61** | **-0.76** |
| 2 | $R^2$ | 0.63 | **0.63** | **0.38** | **0.58** |
| 3 | Trend/unit AOD | | **-470** | **-118** | **-99** |
| 4 | Change over AOD range | -4.29 | **-34.32** | **-18.99** | **-28.70** |





| H | Convection Nino 3.4 Area | | | | |
|---|---|---|---|---|---|
| 1 | R | -0.99 | **-0.82** | **-0.54** | **-0.84** |
| 2 | $R^2$ | 0.99 | **0.68** | **0.29** | **0.70** |
| 3 | Trend/unit AOD | | **-1.12** | **-0.14** | **-0.16** |
| 4 | Change over AOD range | -0.006 | **-0.08** | **-0.02** | **-0.05** |
| I | IOD (May to October) | | | | |
| 1 | R | 0.84 | **0.86** | **0.49** | *0.54* |
| 2 | $R^2$ | 0.70 | **0.73** | **0.24** | *0.29* |
| 3 | Trend/unit AOD | | **57.9** | **3.32** | **2.57** |
| 4 | Change over AOD range | 0.26 | **5.85** | **0.80** | **0.75** |
| J | Pressure SE Australia (Apr-Oct) | | | | |
| 1 | R | 0.98 | **0.74** | **0.49** | **0.70** |
| 2 | $R^2$ | 0.96 | **0.55** | **0.24** | **0.49** |
| 3 | Trend/unit AOD | | **64.03** | **12.82** | **13.95** |
| 4 | Change over AOD range | 0.91 | **5.63** | **2.59** | **4.06** |
| K | Rainfall SE Australia (Apr-Oct) | | | | |
| 1 | R | -0.98 | **-0.62** | **-0.48** | **-0.68** |
| 2 | $R^2$ | 0.96 | **0.39** | **0.23** | **0.46** |
| 3 | Change over AOD range | -73 | **-256** | **-162** | **-78** |
| 4 | % age fall over AOD range | -18% | **-100%** | **-35%** | **-47%** |

Table 2: Correlations and changes from: minimum to maximum tephra/AOD and trend/unit AOD; for the parameters shown in the GVP, LME, M2 and TN datasets. Correlations are all at significance <0.01 (bold) except M2 SEAP Area TS and TN (SON) IOD which show significance <0.02 (italic).



# 4 DISCUSSION

The results show: increasing levels of tephra and AOD in the SEAP Area result in the same change in the parameters analysed; and statistically significant correlations in all cases which clearly establishes the link between tephra/aerosols and ENSO and the associated events.

## 4.1 Causation Analysis

It is clearly understood that correlation between events A and B does not prove causation from A to B or vice versa. Thus,
the causal relationship between the SEAP and the ENSO must be demonstrated in other ways.

## 4.2 Volcanic eruptions

The similarity of the power spectra of the Nino 3.4 SST and the SEAP Area tephra (Fig. 3); and the clear correlations between volcanic tephra in the SEAP Area, ENSO and the other associated events demonstrated in this paper show a close relationship exists.

**Crucially, volcanic eruptions are caused by deep earth tectonic processes and cannot be caused by ENSO or other surface events. Therefore, the causal direction must run from the volcanic eruptions to ENSO and the associated events.**

## 4.3 Immediacy with simple physics

All the data shows that the effects of volcanic tephra and anthropogenic aerosols on ENSO, and other events are immediate
with a simple physical explanation.

## 4.4 Analysis without correlation

Table 2 and Fig. 7 show that the LME Nino 3.4 SST rises by 5.11°C from 297.89°C to 303.00°C and the SOI falls from +8.05 to -26.27 accurately reflecting the range of these indices from La Nina to El Nino events without using correlation. Also, the TN (SON) data shows a much greater variation than the TN(A) data reflecting the much greater variation in the
measured AOD and explaining the ASON anomaly in Fig. 4.

## 4.5 Modelling

LME: the LME data used in this paper is forced by eight agents and, in this modelling, there is no mechanism to create aerosols in south east Asia during an ENSO event and hence the causal direction must run from the aerosols to the ENSO events.





In addition, the aerosol forcings in all LME runs are fixed at 1850 values except for the "ozone and aerosol" and "all" runs and there can therefore be no forcing of the aerosols by any agent within these six runs and the causal direction must flow from the aerosols to the ENSO events.

MERRA-2: The M2 reanalysis assimilates measured aerosol data and as Appendices 1 and 2 show the aerosol sources are volcanoes, gas flares and fires lit by the local population, the causal direction must be from the aerosols to the ENSO events

as source of the aerosols is known and they are assimilated and not generated within the model by ENSO.

### 4.6    Satellite data and recent changes in the SEAP

Figure 4 clearly shows that the character of ENSO as measured by the Nino 3.4 SST has changed since 1980 when the extreme anthropogenic SEAP started to appear and with the high correlations of the anthropogenic SEAP in SON with ENSO it is clear that the change is driven by the SEAP and since Appendix B shows that the SON anthropogenic SEAP is

caused by fires deliberately lit to clear land and agricultural waste the causal direction must run from the SEAP to ENSO.

### 4.7    Seasonality of ENSO

ENSO is highly seasonal and this paper provides an explanation for the seasonality. Rainfall in the CSEAP area and the Nino 3.4 SST correlate at -0.57 significance <0.15 with the SST reducing when rainfall in the SEAP Area is high. This clearly supports the hypothesis that the SEAP is the major cause of ENSO events as the south east Asian monsoon rainfall washes

the aerosols out of the atmosphere enabling convection to be re-established in the region to drive the TWs and allow the Pacific warm pool to relax back to the western Pacific Ocean and end the ENSO event.

### 4.8    Multiple Independent Datasets

Seven of eight LME modelling runs (excluding the aerosol forced run as it correlates with the All forcing run) and the M2 reanalysis exhibit very low or negative correlations between the CSEAP AODVIS in the individual runs as shown in the

correlation matrix in Appendix D with an overall average 0.0016. Hence the datasets are independent.

All these seven LME and M2 datasets show correlations with the ENSO indices at significance of <0.01 or less and the chance that all these eight independent datasets show the same result and are wrong is the product of the significances i.e., $0.01^8$ or $10^{-17}$, a vanishingly small number.

### 4.9    Rainfall and pressure south eastern Australia

Reduced rainfall and increased pressure in SEAus cannot create aerosols in the SEAP Area. I show that drought in SEAus which has commonly been attributed to ENSO and/or IOD events is created simultaneously by the SEAP which further confirms the causal relationship flows from the SEAP to ENSO and the associated events.





## 4.10   ENSO Theory

Two competing ENSO theories exist and this paper, which is a specific case of the steady state with high frequency forcing theory, confirms this theory by clearly showing that the "high frequency" forcing has always been caused by SEAP Area volcanic eruptions and therefore shows the "oscillator" theories to be invalid.

## 4.11   Causal direction

Therefore with:

1.      Volcanic aerosols demonstrating conclusively that SEAP Area aerosols must be the cause of ENSO events;

2.      The LME, M2 and TN data showing a clear connection between the SEAP and ENSO indices which mirrors reality without correlation;

3.      The individual LME, M2 and TN time series analyses showing the same results with a vanishingly small chance of error;

4.      The IOD showing a clear connection to the SEAP; and

5.      Higher pressure and drought in SEAus showing a clear connection the SEAP;

The inevitable conclusion is that the SEAP is the driver of ENSO and these associated events.

## 5   FUTURE RESEARCH

To finally confirm these conclusions, a further LME style analyses should be undertaken in which:

1.      An aerosol plume is created in the model which ramps up from the naturally low level in February to reach the same AOD as the extreme SEAP of October 2006 in March, continues at the same level to October and ramps down in November to the naturally low level in December. This plume to be applied in the model with random returns from 2 to 10 years to mimic the actual return frequency of ENSO; and

2.      Repeating 1 with reducing levels of AOD from February to October to determine the minimum AOD level over SE Asia which is required to cause an ENSO event.

This analysis will conclusively demonstrate that ENSO events are caused by the SEAP and determine the AOD levels required to do so.

## 6   CONCLUSIONS

The GVP volcanic eruption data (151 years), LME (1.156 years), M2 (41 years) and TN (20 years) all confirm the direct connection between the SEAP and ENSO in multiple independent ways.

Causal analysis showing that the relationship must flow from the SEAP to ENSO.





I therefore conclude that the SEAP is unique and is the sole trigger and sustaining agent for ENSO events.

This conclusion brings four important elements into climate change analysis:

**First**: **All Volcanic tephra** from all eruptions and not just sulphate aerosols from large eruptions must be included in climate modelling as they are the prime cause of ENSO events and are therefore the primary interannual climate forcing agent;

**Second**: **Aerosol Regional Dimming**, the surface radiative forcing caused by the annual apparitions of the eight major continental scale, aerosol plumes identified in this paper which now occur each year, the anthropogenic elements of which did not exist before the middle of the 20th century, must be incorporated in climate models at temporal and geographic resolutions which will adequately model their effects. Global, seasonal, annual and decadal averages are insufficient as the climate forcing effects of the plumes only exist when the plume exists and the averaging process reduces the intensity of the plumes and destroys their seasonal effects which, as this paper shows, include the intensification of ENSO events and the perturbation of the major atmospheric circulation systems, the Hadley and Walker Cells.

**Note:** The residence time of aerosols in the troposphere is short – the IPCC AR5 (Stocker et al., 2013) suggested a period of 1 to 3 weeks for volcanic aerosols.

**Third**: **ENSO** Since the SEAP causes ENSO events it follows that this aerosol plume causes an increase in the global temperature, most likely through the modification of the large-scale atmospheric circulation systems (especially convection in SE Asia) instead of just cooling the region under the plume through direct surface forcing as is commonly assumed. It is also likely that the increases in the AOD of the SEAP since 1980 in non-extreme years such as 1999 and 2001 will also have affected the global temperature. This requires further investigation.

**Fourth**: **Multiple Plumes**: The effects of combinations of the eight anthropogenic, continental scale aerosol plumes require investigation as the combined effects of such plumes may be radically different from the effects of individual plumes and this may, for example, provide an explanation for the ridiculously resilient ridge of high pressure in the north east Pacific Ocean which has affected rainfall in western north America in recent years.

Finally I concur with Booth et al. (2012) that emissions of carbonaceous aerosols are directly addressable by government policy actions and suggest that this is an urgent necessity to mitigate future severe anthropogenic ENSO events in the Austral spring which Timbal and Drosdowsky (2013) link to drought in Australia.

## 7    Code/Data Availability

No code is required to recreate the data sets used which are all publicly available and are shown in section 2.3.

## 8    Competing Interests

The author declares that he has no conflict of interest.



## 9    Author Contribution

There are no co-authors.

## 10    Acknowledgments

I acknowledge:

The Global Volcanism Program at the Smithsonian Institution for the volcano eruption data at  https://volcano.si.edu/

NASA: Analyses and visualizations used in this paper were produced with the Giovanni online data system, developed and
maintained by the NASA GES DISC;

The mission scientists and Principal Investigators who provided the data used in this paper including the
CALIPSO data;

Dr Robert Schmunk for the Panoply data viewer;

Earth Observatory for the image of Sangeang Volcano;

The Indonesian burned areas data;

BC Emissions for the Equatorial Asian Region;

Indonesian fire numbers

And Oak Ridge National Laboratory for the Global Fire Emissions Database (GFED4.1s)

The Hadley Centre for the HadISST_1 dataset;

Google Earth™ and the copyright holders noted for the image of the Earth and gas flares;

The Global Forest Watch for fire data;

The Climate and Global Dynamics Division of ESSL at NCAR for the NCEP/NCAR reanalysis SEAP SST, Omega and
Niño 1+2 and 3.4 SST, trade wind and SEAus pressure data;

NOAA: for the ENSO and IOD data at https://psl.noaa.gov/gcos_wgsp/Timeseries/; the trade wind and gas flare data; and
for the data and images from https://psl.noaa.gov/data/gridded/reanalysis/

University Corporation for Atmospheric Research for the LME and HadlSST_1 ENSO return frequency graphs at
http://webext.cgd.ucar.edu/Multi-Case/CVDP_repository/cesm1.lm/nino34.powspec.png

The CESM1(CAM5) Last Millennium Ensemble Community Project and supercomputing resources provided by
NSF/CISL/Yellowstone;

The Volcano Global Risk Identification and Analysis Project for the Toba data

The United Nations Department of Economic and Social Affairs Population Division for the world population statistics;

The Australian Bureau of Meteorology for: the ENSO and Walker Circulation images; SOI data; SOI formula at
http://www.bom.gov.au; and Melbourne rainfall and pressure data;

The IPCC for their assessment reports;

The U.S. Geological Survey for the earthquake information;



BP https://www.bp.com/en/global/corporate/energy-economics/statistical-review-of-world-energy.html for the oil production statistics;

The Global Gas Flaring Reduction Partnership for the gas flaring data






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
