# Peer review of "How extreme apparitions of the volcanic and anthropogenic south east Asian aerosol plume trigger and sustain: El Niño and Indian Ocean Dipole events; and drought in south eastern Australia. First attribution and mechanism using Global Volcanism Program, Last Millennium Ensemble, MERRA-2 reanalysis a"

_Atmospheric Chemistry and Physics, 2022_

## Author Comment (AC1)

**Authors Response to Reviewer 1 Comments**

**My response in BLUE**

Comments on "How extreme apparitions of the volcanic and anthropogenic south east Asian aerosol plume trigger and sustain: El Niño and Indian Ocean Dipole events; and drought in south eastern Australia. First attribution and mechanism using Global Volcanism Program, Last Millennium Ensemble, MERRA-2 reanalysis and NASA satellite data" by Potts et al. Just "Potts" – No "et al"

Before I respond to the Reviewer's comments, I ask the Reviewer and Journal to note that:

The identification of the cause of ENSO and IOD events and thus drought in south eastern Australia, where I live, is a serious social and economic issue as well as a scientific one. Drought in this region especially from September to November:

- Decimates the grain crop. The 2006 drought resulted in a reduction of 67% with a loss of over AUS$2 billion in farmers' income; which
- Resulted in an increase in the rural suicide rate; and
- Has been identified as a cause of the increase in wildfires in the summer; which
- Cause significant loss of life and property

Therefore, the publication of this paper is a vital step in starting the process of reducing the anthropogenic aerosol levels over south east Asia, especially during the September to November "burning season," which will ultimately lead to a mitigation of such droughts and wildfires in south eastern Australia.

The analysis presented in the manuscript is weak. There are too many results without the support of the analysis. The conclusions made in the manuscript are based on the correlation among data sets and past literature. Many statements in the manuscript are overstated. The manuscript in the current form is not suitable for publication in Atmospheric Chemistry and Physics.

I demonstrate that the cause of ENSO, the greatest interannual variation in the climate system, and the IOD has always been the volcanoes in south east Asia, and that in recent decades the effects of these natural aerosols have been intensified by anthropogenic aerosols from September to November. There is no way of presenting these results unless I clearly state their importance as a significant discovery.

I also note that researchers have been trying for decades to show that volcanic eruptions cause ENSO events, and, as I show in my paper, they do, but instead of confining the analysis to large eruptions anywhere in the World, my focus is on all eruptions in south east Asia, a region which is uniquely positioned to simultaneously affect the Walker Circulation, regional Hadley Circulation, the Indian Ocean Dipole and drought and high pressure in south eastern Australia, and does.

I do not understand the questioning of my reliance on "past literature". Surely "past literature" is the basis of peer review as new research builds on the foundation of past research.

If the Reviewer can expand on the "analysis" required I will happily provide it if I can. However, I would point out that all the events I analyse, ENSO, IOD and drought in south eastern Australia are linked in the literature and it is surely axiomatic that a paper which links all of these events to the SEAP contains strong analysis because of this.

Major comments:

I suggest the author perform the analysis using ESM simulations. Using ESM simulations, past studies have shown droughts over Africa and India, are linked to a volcanic aerosol plume and ENSO events. The mechanism involves aerosols causing the reflection of solar radiation leading to cooling at the surface under the plume. It eventually leads to Kelvin wave dissipation in the Central/Eastern Pacific (Khodri, M. et al., Nat. Commun. 2017, Fadnavis S. et al., Scientific Reports, 2021). The author reports a similar mechanism for drought in the south-eastern Australia region. However, the consequences of volcanic aerosols and the proposed mechanism are based on the correlation among data sets. The atmospheric processes are interlinked hence correlation is a weak tool to show the consequences of volcanic aerosols. To show changes in rainfall, temperature, circulation, etc., caused by volcanic aerosols, the author should perform ESM simulations with volcanoes and without volcanoes.

The climate model used for the Last Millennium Ensemble project is described in the reference article, Otto-Bliesner et al. (2016), as:

> "Community Earth System Model-Last Millennium Ensemble (CESM-LME) expands on the CMIP5 and earlier LM model simulations by providing the largest ensemble of LM simulations with a single model to date.";

which shows the LME data I use in the paper is from an Earth System Model (ESM).

In the revised paper I will change the LME plots to show the individual LME simulations from: 850 forcing; Ozone and Aerosols; Greenhouse Gases; Land Use; Orbital, Solar and Volcanic together with one simulation with all these forcings included. There will therefore be one simulation with volcanic forcing and seven without.

Re the two papers mentioned, I will include references to them in the introduction in the revised paper. However, I would state that the explanation for the occurrence of ENSO events in both papers is complex requiring the forcing of Kelvin waves in the Pacific by volcanic eruptions which are remote in space and time from the Pacific and the region which drives the Walker Circulation, whilst my paper shows much higher statistical connections between the SEAP Area volcanoes, located in the region which drives the Walker Circulation, and ENSO. In addition, the connections in my paper are explained by a simple physical model.

The 'results' section is weak. It needs to be strengthened.

The results section in the revised paper will include the change in the LME ESM data presentation mentioned above and the maps which are mainly from the NCEP/NCAR Reanalysis and show the changes from one extremely high SEAP AOD year to one extremely low one will be replaced with maps which use the entire LME or MERRA-2 datasets.

A schematic depicting the processes involved in connecting SEAP and ENSO will be useful.

There are many images showing El Niño and La Niña. This image from the NOAA website some years ago is one of the clearest I have found and I will provide two images: the first of "normal circulation"; and the second is the same image with my overlays showing the sequence of changes forced by the SEAP which cause ENSO events and link them to the hypothesis in the revised paper.

**Normal Circulation**

[Figure]

**Changes Forced by the SEAP**

[Figure]

The introduction section is lengthy and looks disconnected. Here, the author should give past work, gaps, and the reason for undertaking this study. I suggest the author reduce the length of this section considerably.

The discussion section needs to be re-arranged. Please combine subsections to keep integrity.

I will review the introduction and discussion sections for the revised paper as suggested

Otto-Bliesner, B. L., Brady, E. C., Fasullo, J., Jahn, A., Landrum, L., Stevenson, S., Rosenbloom, N., Mai, A., and Strand, G.: Climate Variability and Change since 850 CE: An Ensemble Approach with

the Community Earth System Model, Bulletin of the American Meteorological Society, 97, 735-754, 10.1175/bams-d-14-00233.1, 2016.